# Gravity without averaging

**Andreas Blommaert*** **and Jorrit Kruthoff†**

Stanford Institute for Theoretical Physics, Stanford University, Stanford, CA 94305

* ablommae@stanford.edu, † kruthoff@stanford.edu

## Abstract

We present aspects of a gravitational theory that interpolates between JT gravity, and a gravity theory with a fixed boundary Hamiltonian. For this, we consider a matrix integral with the insertion of a Gaussian with variance $\sigma^2$, centered around a matrix $H_0$. Tightening the Gaussian renders the matrix integral less random, and ultimately it collapses the ensemble to one Hamiltonian $H_0$. This model provides a concrete setup to study factorization, and what the gravity dual of a single member of the ensemble is. Perturbatively around infinite $\sigma$ we find that the JT gravity dilaton potential is modified, and ultimately the gravity theory goes through a series of phase transitions, corresponding to a proliferation of extra macroscopic holes in the spacetime. A good gravitational description at small values of $\sigma$ remains elusive. Furthermore, we observe that in the Efetov model approach to random matrices, the non-averaged factorizing theory is described by one simple saddle point.

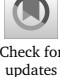

# 1   Introduction

The conventional AdS/CFT correspondence dictates that one single conformal field theory is dual to a single string theory in anti-de Sitter spacetime [1, 2]. A prime example of the correspondence involved $\mathcal{N} = 4$ super Yang-Mills theory in four dimensions on the boundary side and superstring (field) theory on $\text{AdS}_5 \times S^5$ on the bulk side, but there are also examples in other dimensions [2–4]. By now, however, there are also examples in low dimensions where a single bulk theory is not dual to one single boundary theory, but to an ensemble of theories [5–28]. The most notable, is the duality between JT gravity in two dimensions and a certain random matrix ensemble [5].

In recent years, it has been a puzzle how to reconcile these two seemingly different incarnations of the AdS/CFT correspondence. In particular, what the role of averaging is, and what the bulk dual of a single member of the ensemble is. It is important to emphasize that the older examples of AdS/CFT are derived from string theory and, in principle, those examples are UV complete; unlike theories like JT gravity, which are not, but see for instance [29] for a recent attempt to embed JT in string theory. We also want to mention the recent works [30–32] for a fascinating and extremely concrete example within $\text{AdS}_3/\text{CFT}_2$, where some of these questions were addressed. Note as well that in higher dimensions, there are only a few marginal and relevant couplings one could imagine averaging over, if one wants to.

One way of thinking about this puzzle, is that the averaging is just a reflection of the ignorance of UV physics, and in particular JT gravity can only be used to compute self-averaging quantities reliably. Another, compatible, perspective is that when one considers a UV complete theory of quantum gravity, the UV details of the theory, such as branes, strings, higher-spin fields etcetera, get encoded in specific couplings of the effective low energy bulk description. For example this could result in JT gravity with many specific couplings turned on.

The idea would be that this second type of theory is a more realistic toy model of quantum gravity, analogous to selecting one member of the ensemble. In the present paper we investigate in more detail what such a theory would look like. In other words we focus on the question, what is the gravity dual of one single member of the ensemble?

To make progress on that question we consider a deformation of random matrix ensembles (with a known gravitational interpretation) and insert a Gaussian with variance $\sigma$ centered around a *target* Hamiltonian $\mathsf{H}_0$. Upon tightening the Gaussian, the matrix integral localizes around $\mathsf{H}_0$, and at $\sigma = 0$ we have picked out $\mathsf{H}_0$ from the matrix ensemble. By an appropriate double scaling of this theory, we study the effects of the insertion of such a Gaussian on JT gravity.

## 1.1 Less random matrices

More concretely, the matrix model we consider is,[1]

$$\mathcal{Z}(\sigma, \mathsf{H}_0) = \int dH \, \exp\left(-L \operatorname{Tr} V(H) - \frac{L}{2\sigma^2} \operatorname{Tr}(\mathsf{H}_0 - H)^2\right). \tag{1}$$

This integral interpolates between the original matrix ensemble at $\sigma = \infty$

$$\mathcal{Z}(\infty, \mathsf{H}_0) = \int dH \, \exp\left(-L \operatorname{Tr} V(H)\right), \tag{2}$$

and the system with Hamiltonian $\mathsf{H}_0$ - which will be referred to as the target Hamiltonian - when $\sigma = 0$

$$\mathcal{Z}(0, \mathsf{H}_0) = \int dH \, \delta(H - \mathsf{H}_0) \exp\left(-L \operatorname{Tr} V(H)\right). \tag{3}$$

This follows since appropriately normalized tight Gaussians are distributionally identical to Dirac deltas,

$$\lim_{\sigma \to 0} \left(\frac{L}{2\pi\sigma^2}\right)^{L(L+1)/4} \exp\left(-\frac{L}{2\sigma^2} \operatorname{Tr}(H - \mathsf{H}_0)^2\right) = \delta(H - \mathsf{H}_0). \tag{4}$$

Morally, the point is that random matrices are less random when their potential is very sharply peaked around some target $\mathsf{H}_0$ eigenvalues; the uncertainly on the position of the random eigenvalues is strongly reduced, because a strong external force is attracting them to the target $\mathsf{H}_0$ eigenvalues.

Expanding out (1), and dropping an overall constant that cancels in all observables, one obtains,

$$\mathcal{Z}(\sigma, \mathsf{H}_0) = \int dH \, \exp\left(-L \operatorname{Tr}\left(V(H) + \frac{1}{2\sigma^2} H^2\right) - \frac{L}{\sigma^2} \operatorname{Tr}(\mathsf{H}_0 H)\right). \tag{5}$$

This is just a matrix integral with potential $V(H) + H^2/2\sigma^2$ coupled to an external field $\mathsf{H}_0$, which has been studied extensively in the literature [33–51].

The question now is, what is the two dimensional quantum gravity of these models? Famously, minimal string theories are obtained by double scaling finite dimensional matrix integrals near the spectral edge $E_0$ of the leading order density of states [52–54] - the double scaling procedure involves sending $E_0$ to infinity and simultaneously sending $L$ to infinity in such a way that the spectral density near the edge, remains finite.[2] JT gravity can be obtained as a further $p \to \infty$ limit of the $(2, p)$ minimal strings [5, 56–59], and concordantly is also a double scaled matrix integral [5].

Except for providing a remarkably complete matching between all genus amplitudes – not previously achieved for minimal strings, due to a lack of precise formulas for conformal blocks – perhaps the most important insight in [5] was the realization that the random matrix $H$ should literally be interpreted as the Hamiltonian of the boundary dual to JT dilaton gravity. This invites to instead view the minimal string theories as bona fide theories of 2d quantum gravity, interpreting the worldsheets as spacetimes. Remarkably they too can be rewritten as dilaton gravities, and the random matrix again gains physical significance as the random

---

[1] Note that we can multiply this with arbitrary overall normalization constants whenever we see fit, these cancel out in all observables.

[2] Equivalently, in the older matrix model literature one would send $L \to \infty$ and tuning simultaneously to the critical point of the matrix model [55].

Hamiltonian [56, 57]; not just an abstract field in a nonperturbative definition of quantum gravity.

This gives immediate gravitational motivation for considering our model (1). Our goal is to understand this theory (1) for finite values of $\sigma$, and follow as much as possible how it transitions from random to non-random.

## 1.2 Summary, structure and main lessons

The summary and structure of the rest of the paper is as follows.

We start by investigating the simplest possible example, the finite dimensional Gaussian matrix integral. By using techniques of [36, 37, 40] we can exactly compute the spectrum, and spectral correlation, for any value of $\sigma$. It is satisfying to visually see this theory transition from completely random to entirely non-random, as summarized in Fig. 2 and Fig. 3.

Ultimately we are interested in continuum gravity, so we try to extract geometric lessons from these exact manipulations. Our main observations are the following:

1. The **wormhole** geometry in the completely averaged theory $\sigma = \infty$, approaches **diagonal** delta functions near the completely fixed theory $\sigma = 0$, as **factorization** requires. This resonates well with earlier discussions about how gravitational systems could factorize [11, 12, 60], see **section 2.2**.

2. Nonperturbative effects in matrix integrals are best captured via another, dual matrix integral known as the Efetov model [61, 62]. It has an interesting saddle point structure that for example explains the plateau via the Andreev-Altshuler instanton [62, 63]. There are new saddle points for finite $\sigma$. These are one to one with solutions of the spectral curve equation

$$y = \frac{4E}{a^2} - \frac{4}{La^2} \sum_{i=1}^{L} \frac{1}{y - x_i/\sigma^2}, \quad \frac{2}{a^2} = \frac{2}{b^2} + \frac{1}{2\sigma^2}, \quad (6)$$

for the Gaussian model with an external field (9) [47]. Here $x_i$ are the eigenvalues of the target Hamiltonian $H_0$. The gravitational interpretation of these new saddles involves D-branes, much like the interpretation of the Andreev-Altshuler saddle itself; this is an invitation to universe field theory [18], where D-brane effects have natural gravity interpretations. Notably, **one universal saddle point $S = 0$ governs the non-random theory** at $\sigma = 0$. See **section 2.3**.

3. At small $\sigma$ the matrix integrals develops $L$ narrow cuts with only one eigenvalue in each of them, on average. Zooming in on a tight semicircle, there still is random matrix universality, however there are deserts between these tiny cuts where the spectrum and spectral correlations essentially vanish. The theory becomes less-random because of these deserts. See **section 2.3**.

4. We find a **dispersion relation** for each observable, expressing the completely fixed theory for $\sigma = 0$ as the completely random theory at $\sigma = \infty$, plus non-self averaging contributions associated with **other poles** in the complex $\sigma$ plane. This explains how the averaged geometry is always contained in the non-averaged theory. This is analogous to [60], especially when applying this to the Efetov model. This model can be thought of as the $G\Sigma$ theory of SYK but now for matrix integrals. See **section 2.4**. The gravitational interpretation of the other poles remains largely unclear, see **section 6**.

5. By studying the ribbons graphs, one observes a tendency for huge holes to form when $\sigma$ becomes small, see **section 2.5**. In gravity this is the tearing of spacetime observed in [64], see **section 4**.

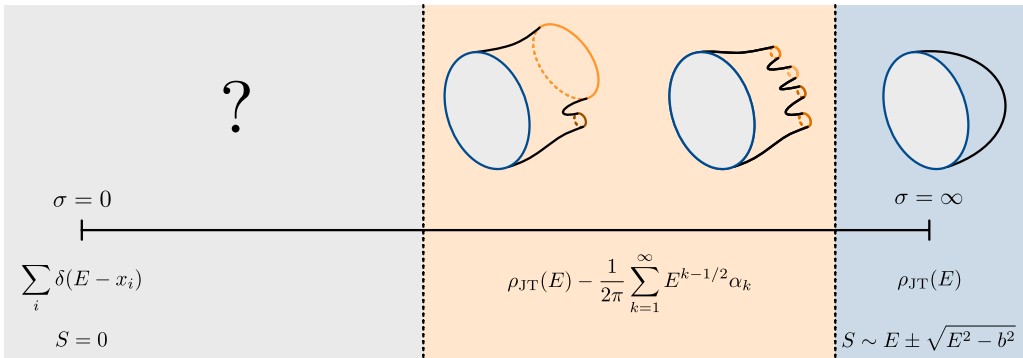

Figure 1: Phase diagram of the matrix integral (1) and its gravitational interpretation as a function of $\sigma$. On the far right (blue region), we have $\sigma = \infty$ and our matrix model is that for JT gravity. We also added the saddle for the Efetov sigma model in the Gaussian model at $\sigma = \infty$. As we move away from $\sigma = \infty$ we enter in the orange region, where the matrix model is deformed by (ghost) brane insertions that are labelled by the eigenvalues $x_i$ of the target Hamiltonian $\mathsf{H}_0$ (the different shades of orange on the small boundaries is supposed to represent that). The spectral density is given as well in this region with $\alpha_k$ given in 108. As $\sigma$ is decreased further in the orange region the model goes through a series of *tearing* phase transitions [64], which is manifested geometrically by very large boundaries ending on the brane. Decreasing $\sigma$ further results in the breakdown of various approximations we had made in section 3 and the model seems to enter a branched polymer phase, see section 6. At $\sigma = 0$ the theory is completely fixed to $H = \mathsf{H}_0$. Remarkably, the Efetov model localizes on one universal saddle $S = 0$ in this regime.

Next we are interested in investigating how the JT dilaton gravity path integral changes upon tuning the Hamiltonians from random matrices (2) to non-random matrices (3). The endgame is capturing what gravitational systems with a single boundary dual might look like and which new ingredients can appear. Therefore we will study (1) in the double scaling limit focusing on three regimes. We obtain the following main results (see also Fig. 1):

1. For large $\sigma$, the external matrix $\mathsf{H}_0$ results in a **deformation** of the JT gravity **dilaton potential** of the type discussed in [65, 66], and applied later for example in [57, 67, 68]

$$I = -\frac{1}{2} \int d^2 x \sqrt{g} \left[ \Phi (R + 2) + 2 U(\Phi, \sigma, \mathsf{H}_0) \right] . \tag{7}$$

   The explicit expression for the dilaton potential as function of $\sigma$ and the eigenvalues of the target Hamiltonian $\mathsf{H}_0$ is presented in (116). This can be viewed as JT gravity with many local operators inserted. The take away is there are perfectly sensible theories of dilaton gravity which are slightly less random than JT gravity, see **section 3**.

2. We investigate how the system transitions from large to small $\sigma$, and find that a phase transition in the matrix integral occurs, structurally similar to the one discussed in [64]. On the far side of the transition, the spacetimes are utterly destroyed by the proliferation of huge holes, we call this the **tearing phase** of gravity, following [64], see **section 4**.

3. We explain that fixing the whole Hamiltonian $H$ is overkill if one is interested only in factorization [5, 8, 10–12]. Instead, one could keep most of the eigenvalues of $H$ random, and only gradually fix some eigenvalues of $H$ towards eigenvalues of $\mathsf{H}_0$, by tuning $\sigma$. In such a scenario, one treats most of the target Hamiltonian $\mathsf{H}_0$ as random, resulting

in a two-matrix integral [69–73], and one fixes some eigenvalues of $H_0$ to $x_1 \ldots x_n$ by considering the partition function

$$\mathcal{Z}(\sigma, x_1 \ldots x_n) = \int \mathrm{d}H \int \mathrm{d}H_0 \, \mathrm{Tr} \, \delta(H_0 - x_1) \ldots \mathrm{Tr} \, \delta(H_0 - x_n) \times$$

$$\times \exp\left( -L \, \mathrm{Tr} \, V(H) - \frac{L}{2\sigma^2} \, \mathrm{Tr}(H_0 - H)^2 \right). \qquad (8)$$

Integrating out the matrix $H_0$ and working at small $\sigma$, this results to leading order in eigenbranes for the matrix $H$ [11], but with a Gaussian **smearing** that is reminiscent of the Gaussian peaks in the finite dimensional matrix integral of section 2. Each of the eigenbranes represents a macroscopic hole in spacetime. Subleading corrections involve ever more macroscopic boundaries, surprisingly. See **section 5**.

We close off in **section 6** with a discussion on lessons for higher-dimensions, the gray region in Fig. 1, and present open questions.

## 2 Gaussian matrix integral

The Gaussian matrix integral coupled to an external matrix $H_0$ is given by (5)

$$\mathcal{Z}(\sigma, H_0) = \int \mathrm{d}H \exp\left( -\frac{2L}{a^2} \, \mathrm{Tr}(H^2) + \frac{L}{\sigma^2} \, \mathrm{Tr}(H_0 H) \right), \quad \frac{2}{a^2} = \frac{2}{b^2} + \frac{1}{2\sigma^2}. \qquad (9)$$

Here $b$ indicates the edge of the spectrum at large $L$ for the undeformed theory [62, 74], this gets shifted due to the Gaussian deformation. We introduced $2/a^2$ to compactify some formulas down the road.

We would like to compute the spectrum of this theory

$$\rho(E) = \mathrm{Tr} \, \delta(E - H), \qquad (10)$$

and moments of the spectrum, like the spectral correlation $\rho(E_1)\rho(E_2)$, which measures the correlation between different eigenvalues of $H$. We achieve this by first computing the real-time partition

$$Z(\mathrm{i}t) = \mathrm{Tr}\left(e^{\mathrm{i}tH}\right), \qquad (11)$$

or, more particularly, the ensemble average $\langle Z(it) \rangle$ with the probability distribution given in (9); and then Fourier transforming to obtain spectral correlators, for example:

$$\langle \rho(E) \rangle = \langle \mathrm{Tr} \, \delta(E - H) \rangle = \int_{-\infty}^{+\infty} \frac{\mathrm{d}t}{2\pi} e^{-\mathrm{i}Et} \langle Z(\mathrm{i}t) \rangle. \qquad (12)$$

One novelty is that the term $\mathrm{Tr}(H_0 H)$ breaks the $U(L)$ invariance of standard matrix integrals. When we go to an eigenvalue basis for the random Hamiltonians

$$H = U \Lambda U^\dagger, \quad \Lambda = \mathrm{diag}(\lambda_1, \ldots, \lambda_L), \qquad (13)$$

the integral over Haar random unitaries $U$ does not decouple, and we need to explicitly compute that integral too. Here we are interested in computing expectations values of only $U(L)$ invariant observables like (10) or (11). In these cases, fortunately, the unitary integral can be done exactly using a beautiful result by Harish-Chandra and - decades later - by physicists Itzykson and Zuber [75, 76].

To see how this works, let's consider some generic $U(L)$ invariant observable $F(H)$. Diagonalizing $H$, and including the Jacobian [74] this becomes

$$\langle F(H)\rangle = \frac{1}{\mathcal{Z}(\sigma, \mathsf{H}_0)} \int_{-\infty}^{+\infty} \prod_{i=1}^{L} d\lambda_i \, \exp\left(-\frac{2L}{a^2}\sum_{i=1}^{L}\lambda_i^2\right)\Delta(\lambda)^2 F(\lambda)\times$$
$$\times \int dU \, \exp\left(\frac{L}{\sigma^2}\mathrm{Tr}\left(\mathsf{H}_0 \, U\,\Lambda\,U^\dagger\right)\right), \tag{14}$$

which features the famous Vandermonde determinant

$$\Delta(\lambda) = \prod_{i<j}^{L}(\lambda_i - \lambda_j). \tag{15}$$

The unitary integral is the sole modification to the matrix model as compared to the undeformed case. This can be computed using the Harich-Chandra formula [75,76]

$$\int dU \, \exp\left(\frac{L}{\sigma^2}\mathrm{Tr}\left(\mathsf{H}_0\,U\,\Lambda\,U^\dagger\right)\right) = \left(\frac{\sigma^2}{L}\right)^{L(L-1)/2}\prod_{n=1}^{L-1} n! \, \frac{1}{\Delta(x)\Delta(\lambda)}\,\det\left(\exp\left(\frac{L}{\sigma^2}x_i\lambda_j\right)\right). \tag{16}$$

Here $x_i$ are the eigenvalues of $\mathsf{H}_0$. Using more modern techniques one derives this by noticing that this unitary group integral is one-loop exact and the Duistermaat-Heckman theorem applies [77,78].

Extracting the constant prefactor from the partition function (9) and using the symmetries of the integrand under exchanging eigenvalues, one obtains

$$\langle F(H)\rangle = \frac{1}{\mathcal{Z}(\sigma, \mathsf{H}_0)} \int_{-\infty}^{+\infty} \prod_{i=1}^{L} d\lambda_i \, \exp\left(-\frac{2L}{a^2}\sum_{i=1}^{L}\lambda_i^2 + \frac{L}{\sigma^2}\sum_{i=1}^{L}x_i\lambda_i\right)\frac{\Delta(\lambda)}{\Delta(x)}F(\lambda). \tag{17}$$

The effect of the deformation is thus to change the potential for the eigenvalues in two ways, first by the explicit term in the exponential, coupling the eigenvalues of $H$ to those of $\mathsf{H}_0$, and second by replacing $\Delta(\lambda)^2$ with $\Delta(\lambda)/\Delta(x)$. Despite appearances perhaps, there is still quadratic level repulsion [62,74] in this ensemble; the cluster function $T(E_1, E_2)$ retains a quadratic maximum, see Fig. 3.

Notice that this trivially extends to other potentials $V(H)$ instead of the quadratic Gaussian.

Now we only need to work out the eigenvalue integral. In the Gaussian case this is straightforward, but nevertheless gives a lot of insight into what changes the coupling to the external matrix $\mathsf{H}_0$ causes. One major change is that the spectral density should interpolate between a sum of delta functions and the semi-circle. Another important change is that as $\sigma$ becomes small, spectral correlation becomes smaller. To show this, we now compute the spectral density and spectral correlation. See also [36,40].

## 2.1 Spectrum

As announced we first compute the real-time partition function, which from (17) becomes

$$\langle Z(it)\rangle = \frac{1}{\mathcal{Z}(\sigma, \mathsf{H}_0)}\int_{-\infty}^{+\infty}\prod_{i=1}^{L} d\lambda_i \, \exp\left(-\frac{2L}{a^2}\sum_{i=1}^{L}\lambda_i^2 + \frac{L}{\sigma^2}\sum_{i=1}^{L}x_i\lambda_i\right)\frac{\Delta(\lambda)}{\Delta(x)}\sum_{j=1}^{L}e^{it\lambda_j}. \tag{18}$$

The integral over the eigenvalues $\lambda_i$ can be done explicitly using the result

$$\int_{-\infty}^{+\infty}\prod_{i=1}^{L} d\lambda_i \, \exp\left(-\frac{2L}{a^2}\sum_{i=1}^{L}\lambda_i^2 + L\sum_{i=1}^{L}q_i\lambda_i\right)\Delta(\lambda) \propto \exp\left(\frac{La^2}{8}\sum_{i=1}^{L}q_i^2\right)\Delta(q). \tag{19}$$

The normalization constant drops out, when we do the same integral for the denominator of (18). In our calculation $q_i = x_i/\sigma^2 + \mathrm{i}t\delta_{ij}/L$, so that the ratio of Vandermonde determinants $\Delta(q)/\Delta(x)$ becomes

$$\frac{\Delta(q)}{\Delta(x)} \propto \prod_{i<k}^{L} \frac{x_i - x_k + \mathrm{i}t\sigma^2\delta_{ij}/L - \mathrm{i}t\sigma^2\delta_{kj}/L}{x_i - x_k} = \prod_{p\neq j}^{L} \left(1 + \frac{\mathrm{i}t\sigma^2/L}{x_p - x_j}\right). \tag{20}$$

The terms in the first product are one except when $i = j$ or $k = j$. Combining everything, including a similar integral for the denominator of (18), one finds

$$\langle Z(\mathrm{i}t)\rangle = \sum_{j=1}^{L} \exp\left(\mathrm{i}tx_j \frac{a^2}{4\sigma^2} - t^2\frac{a^2}{8L}\right) \prod_{i\neq j}^{L} \left(1 + \frac{\mathrm{i}t\sigma^2/L}{x_i - x_j}\right). \tag{21}$$

As consistency check, the normalization works out because $\langle Z(0)\rangle = L$.

This looks rather unpleasant for manipulations, due to the sum and product. This improves when exchanging the sum over $j$ for a contour integral around all eigenvalues $x_j$ of $\mathsf{H}_0$

$$\langle Z(\mathrm{i}t)\rangle = \frac{L}{\mathrm{i}t\sigma^2} \oint_{\mathsf{H}_0} \frac{\mathrm{d}u}{2\pi\mathrm{i}} \prod_{i=1}^{L} \left(1 + \frac{\mathrm{i}t\sigma^2/L}{u - x_i}\right) \exp\left(-\frac{\sigma^2}{2L}t^2\frac{1}{1 + 4\sigma^2/b^2} + \mathrm{i}tu\frac{1}{1 + 4\sigma^2/b^2}\right), \tag{22}$$

where the subscript $\mathsf{H}_0$ on the contour integral indicates the collection of small contours around each eigenvalue $x_i$ of $\mathsf{H}_0$. Each pole generates one term in the sum. The spectral density is then the Fourier transform of (22)

$$\langle \rho(E)\rangle = \int_{-\infty}^{\infty} \frac{\mathrm{d}t}{2\pi} \frac{L}{\mathrm{i}t\sigma^2} \oint_{\mathsf{H}_0} \frac{\mathrm{d}u}{2\pi\mathrm{i}} \prod_{i=1}^{L} \left(1 + \frac{\mathrm{i}t\sigma^2/L}{u - x_i}\right) \times$$
$$\times \exp\left(-\frac{\sigma^2}{2L}t^2\frac{1}{1 + 4\sigma^2/b^2} + \mathrm{i}t\left(u\frac{1}{1 + 4\sigma^2/b^2} - E\right)\right). \tag{23}$$

In the extremal regimes of $\sigma$ we deduce the following behavior:

1. For $\sigma$ small, we expand the product over $i$. The order $\sigma^0$ term in the product does not contribute, because is has no poles, and thus the leading contribution comes from the $\sigma^2$ term in the product. Furthermore we can approximate the exponent for $\sigma^2 \ll b^2$; in total we then obtain

$$\langle \rho(E)\rangle = \int_{-\infty}^{\infty} \frac{\mathrm{d}t}{2\pi} \oint_{\mathsf{H}_0} \frac{\mathrm{d}u}{2\pi\mathrm{i}} \frac{1}{u - x_i} \exp\left(-\frac{\sigma^2}{2L}t^2 + \mathrm{i}t(u - E)\right) \tag{24}$$

$$= \left(\frac{L}{2\pi\sigma^2}\right)^{1/2} \sum_{i=1}^{L} \exp\left(-\frac{L}{2\sigma^2}(E - x_i)^2\right)$$

$$= \sum_{i=1}^{L} \delta(E - x_i), \tag{25}$$

where the last line uses the definition of Dirac deltas (4) for $\sigma = 0$. This is indeed the expected spectral density for a system with non-random Hamiltonian $\mathsf{H}_0$.

2. for $\sigma$ large we rescale $u \to u\sigma^2$ and expand around large $\sigma$, this effectively pushes all poles towards the origin $u = 0$. The product over $i$ then simplifies and becomes

independent of the eigenvalues of $H_0$. Then we can furthermore enforce the large $L$ limit, using a limit representation of $e^x$

$$
\begin{aligned}
\langle Z(\mathrm{i}t)\rangle &= \frac{L}{\mathrm{i}t} \oint_0 \frac{\mathrm{d}u}{2\pi\mathrm{i}} \left(1 + \frac{1}{L}\frac{\mathrm{i}t}{u}\right)^L \exp\left(-\frac{b^2}{8L}t^2 + \mathrm{i}tu\frac{b^2}{4}\right) \\
&= \frac{2L}{\mathrm{i}tb} \oint_0 \frac{\mathrm{d}u}{2\pi\mathrm{i}} \exp\left(\frac{\mathrm{i}tb}{2}\left(\frac{1}{u} + u\right)\right),
\end{aligned}
\tag{26}
$$

in the second equality we again rescaled $u \to 2u/b$ for convenience. This contour integral can be done by using the generating function of the Bessel functions,

$$
\exp\left(\frac{\mathrm{i}bt}{2}\left(u + \frac{1}{u}\right)\right) = \sum_{k=-\infty}^{+\infty} (\mathrm{i}u)^k J_k(bt).
\tag{27}
$$

The contour integral over $u$ picks out the $k = -1$ term in the sum and we recover the known genus zero partition function of the Gaussian matrix integral [79]

$$
\langle Z(\mathrm{i}t)\rangle = \frac{2L}{tb} J_1(bt).
\tag{28}
$$

Fourier transforming this gives the semicircle [62,74], with implicit Heaviside

$$
\langle \rho(E)\rangle = \frac{2L}{\pi b^2}(b^2 - E^2)^{1/2}.
\tag{29}
$$

To summarize, at large $\sigma$ we obtain the standard result for the undeformed Gaussian matrix integral, whereas for small $\sigma$ we obtain a sum of delta functions. This is of course no surprise, but helps to understand the full result (23).

We are especially interested in understanding how the system transitions from the completely random result (29), to the delta spikes (25). One approach is to expand around both large and small $\sigma$. Those regimes are discussed at length in the double scaled regime, with the corresponding gravitational interpretation, in respectively section 3 and section 5; we choose not to repeat that exercise here.

Instead we exploit the strengths of the finite dimensional theory. It is rewarding to just plot $\rho(E)$ for several values of $\sigma$ and $L = 8$, see Fig. 2. At both extreme values of $\sigma$ we find the expected result, whereas at intermediate values of $\sigma$ the oscillations in the spectral density are large, eventually resulting in regions where the eigenvalue support is exponentially small.

Below in section 2.3 and 2.4 we comment on potential nonperturbative gravitational interpretations associated with these oscillations, and the transition as a whole. An important role seems to be played by extra saddle points in the Efetov model formulation [61,62] of the Gaussian matrix integral and by poles of observables, as function of $\sigma$. We have not yet succeeded in double scaling these particular aspects and consider this an important open problem, see section 6.

However, let us first perform a similar analysis for the spectral correlation, which is relevant for the factorization problem.

## 2.2 Eigenvalue correlation and factorization

To compute the eigenvalue correlation, we first consider the spectral form factor. Using (17), one finds

$$
\begin{aligned}
\langle Z(\mathrm{i}t_1)Z(\mathrm{i}t_2)\rangle = \frac{1}{\mathcal{Z}(\sigma, H_0)} \int_{-\infty}^{+\infty} \prod_{i=1}^L \mathrm{d}\lambda_i \, \exp\left(-\frac{2L}{a^2}\sum_{i=1}^L \lambda_i^2 + \frac{L}{\sigma^2}\sum_{i=1}^L x_i\lambda_i\right) \frac{\Delta(\lambda)}{\Delta(x)} \times \\
\times \sum_{j=1}^L e^{\mathrm{i}t_1\lambda_j} \sum_{k=1}^L e^{\mathrm{i}t_2\lambda_k}
\end{aligned}
$$

$$= \langle Z(i(t_1 + t_2)) \rangle + \frac{1}{\mathcal{Z}(\sigma, \mathsf{H}_0)} \int_{-\infty}^{+\infty} \prod_{i=1}^{L} d\lambda_i \exp\left(-\frac{2L}{a^2} \sum_{i=1}^{L} \lambda_i^2 + \frac{L}{\sigma^2} \sum_{i=1}^{L} x_i \lambda_i\right) \times$$

$$\times \frac{\Delta(\lambda)}{\Delta(x)} \sum_{j \neq k}^{L} e^{it_1\lambda_j + it_2\lambda_k}.$$

Using (19) with $q_i = x_i/\sigma^2 + it_1\delta_{ij}/L + it_2\delta_{ik}/L$ we perform the integral over eigenvalues; the ratio $\Delta(q)/\Delta(x)$ becomes in this case

$$\frac{\Delta(q)}{\Delta(x)} \propto \left(1 + \frac{i(t_1 - t_2)\sigma^2/L}{x_j - x_k}\right) \prod_{i \neq (j,k)}^{L} \left(1 + \frac{it_1\sigma^2/L}{x_j - x_i}\right) \left(1 + \frac{it_2\sigma^2/L}{x_k - x_i}\right). \tag{30}$$

The eigenvalue integral of the off-diagonal $j \neq k$ terms in (30) therefore becomes

$$\sum_{j \neq k}^{L} \exp\left(i(t_1 x_j + t_2 x_k)\frac{a^2}{4\sigma^2} - (t_1^2 + t_2^2)\frac{a^2}{8L}\right) \left(1 + \frac{i(t_1 - t_2)\sigma^2/L}{x_j - x_k}\right) \times$$

$$\times \prod_{i \neq (j,k)}^{L} \left(1 + \frac{it_1\sigma^2/L}{x_j - x_i}\right) \left(1 + \frac{it_2\sigma^2/L}{x_k - x_i}\right).$$

Notice as check that $\langle Z(0)Z(0) \rangle = L^2$. Introducing contour integrals, this is reorganized further into

$$\frac{L}{it_1\sigma^2} \oint_{\mathsf{H}_0} \frac{du_1}{2\pi i} \exp\left(it_1 u_1 \frac{a^2}{4\sigma^2} - t_1^2 \frac{a^2}{8L}\right) \prod_{i=1}^{L} \left(1 + \frac{it_1\sigma^2/L}{u_1 - x_i}\right) \frac{L}{it_2\sigma^2} \times \tag{31}$$

$$\times \oint_{\mathsf{H}_0} \frac{du_2}{2\pi i} \exp\left(it_2 u_2 \frac{a^2}{4\sigma^2} - t_2^2 \frac{a^2}{8L}\right) \prod_{j=1}^{L} \left(1 + \frac{it_2\sigma^2/L}{u_2 - x_j}\right) \times \tag{32}$$

$$\times \frac{(u_1 - u_2 + i(t_1 - t_2)\sigma^2/L)(u_1 - u_2)}{(u_1 - u_2 + it_1\sigma^2/L)(u_1 - u_2 - it_2\sigma^2/L)} = \tag{33}$$

$$= \langle Z(it_1) \rangle \langle Z(it_2) \rangle + \oint_{\mathsf{H}_0} \frac{du_1}{2\pi i} \exp\left(it_1 u_1 \frac{a^2}{4\sigma^2} - t_1^2 \frac{a^2}{8L}\right) \prod_{i=1}^{L} \left(1 + \frac{it_1\sigma^2/L}{u_1 - x_i}\right) \times \tag{34}$$

$$\times \frac{1}{u_1 - u_2 - it_2\sigma^2/L} \oint_{\mathsf{H}_0} \frac{du_2}{2\pi i} \exp\left(it_2 u_2 \frac{a^2}{4\sigma^2} - t_2^2 \frac{a^2}{8L}\right) \times \tag{35}$$

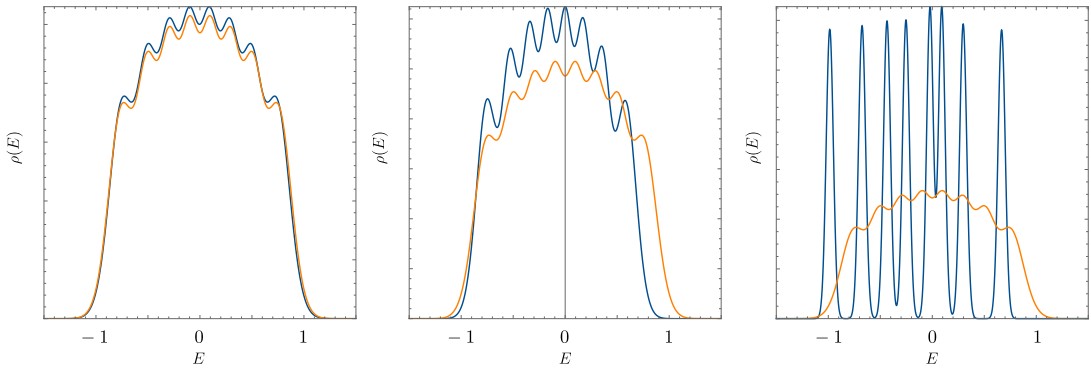

Figure 2: Spectrum $\rho(E)$ for $L = 8$ and $\sigma = 3, 1/2, 1/10$ (left to right); it transitions from the semicircle (orange) to a sum of deltas on the eigenvalues of $\mathsf{H}_0$. For intermediate values of $\sigma$ there are heavy oscillations. The sharper the peaks and valleys in the spectrum, the less random the matrix integral.

$$\times \prod_{j=1}^{L} \left(1 + \frac{\mathrm{i}t_2 \sigma^2/L}{u_2 - x_i}\right) \frac{1}{u_1 - u_2 + \mathrm{i}t_1 \sigma^2/L}.$$

This expression simplifies further when one Fourier transforms to the spectral correlation

$$\langle \rho(E_1)\rho(E_2)\rangle = \int_{-\infty}^{+\infty} \frac{\mathrm{d}t_1}{2\pi} e^{-\mathrm{i}E_1 t_1} \int_{-\infty}^{+\infty} \frac{\mathrm{d}t_2}{2\pi} e^{-\mathrm{i}E_2 t_2} \langle Z(\mathrm{i}t_1)Z(\mathrm{i}t_2)\rangle. \tag{36}$$

Within the double contour integral in (34), we can shift the time variables as $t_1 \to t_1 + \mathrm{i}(u_1 - u_2)L/\sigma^2$ and $t_2 \to t_2 - \mathrm{i}(u_1 - u_2)L/\sigma^2$; this factorizes the double contour integral. Combining this, with the first term in (30) and with the first term in (34), one finally arrives at the elegant answer

$$\langle \rho(E_1)\rho(E_2)\rangle = \delta(E_1 - E_2)K(E_1, E_1) + K(E_1, E_1)K(E_2, E_2) - K(E_1, E_2)K(E_2, E_1), \tag{37}$$

where the all-encompassing kernel is derived to be

$$K(E_1, E_2) = \int_{-\infty}^{\infty} \frac{\mathrm{d}t}{2\pi} \frac{L}{\mathrm{i}t\sigma^2} \oint_{\mathsf{H}_0} \frac{\mathrm{d}u}{2\pi \mathrm{i}} \prod_{i=1}^{L} \left(1 + \frac{\mathrm{i}t\sigma^2/L}{u - x_i}\right) \times$$
$$\times \exp\left(-t^2 \frac{a^2}{8L} + \mathrm{i}t\left(u\frac{a^2}{4\sigma^2} - E_2\right) + u(E_1 - E_2)\frac{L}{\sigma^2}\right). \tag{38}$$

On the diagonal $E_1 = E_2$ this kernel reduces to the spectrum (23). In the usual GUE matrix model, this kernel is an important object, because all spectral correlators can be expressed as sums of products of these kernels [74]. This conclusion extends to the matrix model with an external field (1), the two point function (37) is just the simplest example [33,36,37,40].

Following Mehta [74], we introduce the smooth part of the eigenvalue correlation $R(E_1, E_2)$ and the eigenvalue covariance $T(E_1, E_2)$, using (37) these become

$$R(E_1, E_2) = K(E_1, E_1)K(E_2, E_2) - K(E_1, E_2)K(E_2, E_1),$$
$$T(E_1, E_2) = K(E_1, E_2)K(E_2, E_1). \tag{39}$$

These quantities were plotted for several values of $\sigma$ and $L = 8$ in Fig. 3. These figures are important for understanding how the system gradually achieves factorization, as discussed below.

As before, it is instructive to analyze the extremal regimes of $\sigma$ analytically. We obtain the following behavior:

1. For $\sigma$ small, we expand the product over $i$. The order $\sigma^0$ term in the product does not contribute, because there are no poles, and so the leading contribution comes from the $\sigma^2$ term in the product, just as for the one-point function. Furthermore approximating the exponent for $\sigma^2 \ll b^2$, one obtains

$$K(E_1, E_2) = \int_{-\infty}^{\infty} \frac{\mathrm{d}t}{2\pi} \oint_{\mathsf{H}_0} \frac{\mathrm{d}u}{2\pi \mathrm{i}} \sum_{i=1}^{L} \frac{1}{u - x_i} \exp\left(-\frac{\sigma^2}{2L}t^2 + \mathrm{i}t(u - E_2) + u(E_1 - E_2)\frac{L}{\sigma^2}\right) \tag{40}$$

$$= \left(\frac{L}{2\pi\sigma^2}\right)^{1/2} \sum_{i=1}^{L} \exp\left(-\frac{L}{2\sigma^2}(E_2 - x_i)^2 + \frac{L}{\sigma^2}x_i(E_1 - E_2)\right). \tag{41}$$

The spectral covariance (39) for small $\sigma$ then becomes, after simply inserting the kernels

$$T(E_1, E_2) = \left(\frac{L}{2\pi\sigma^2}\right) \sum_{i,j=1}^{L} \exp\left(-\frac{L}{2\sigma^2}(E_2 - x_i)^2 - \frac{L}{2\sigma^2}(E_1 - x_j)^2 - \frac{L}{\sigma^2}(E_1 - E_2)(x_j - x_i)\right)$$

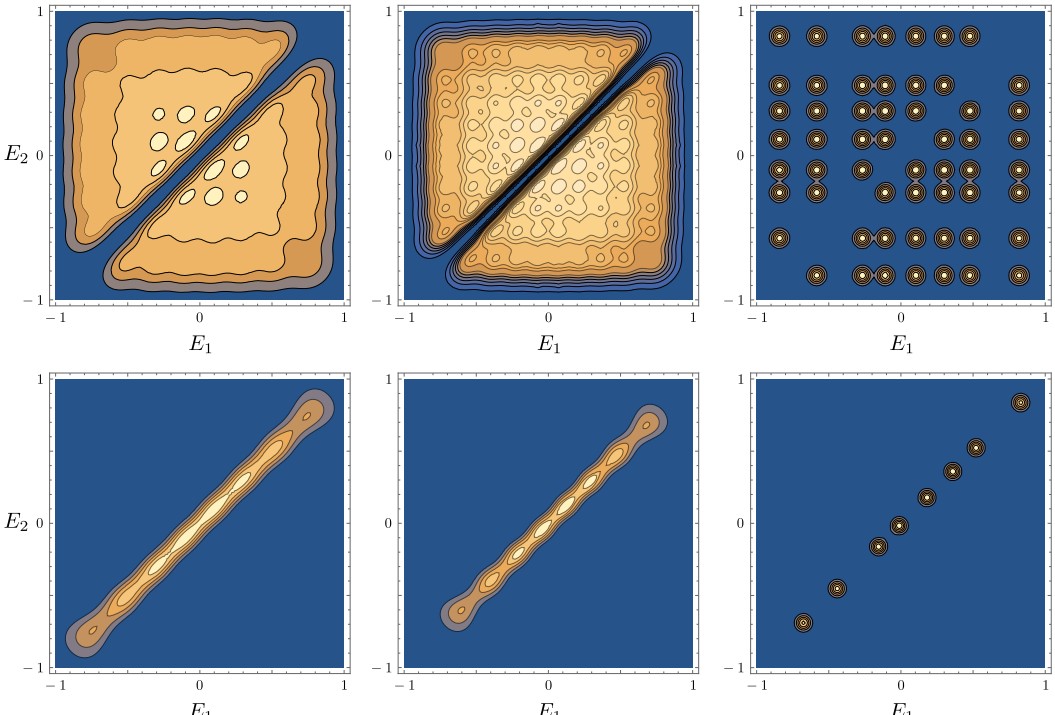

Figure 3: Spectral correlation $R(E_1, E_2)$ (top) and spectral covariance $T(E_1, E_2)$ (bottom) for $L = 8$ and $\sigma = 5, 1, 1/10$ (left to right). The covariance $T(E_1, E_2)$ is only significant close to the diagonal axis, nearby eigenvalues repel; it interpolates between a ridge and the diagonal deltas. The theory factorizes, because the covariance drops to zero everywhere, except on the location of the eigenvalues of $H_0$ where it produces the required delta contact terms. The correlation $R(E_1, E_2)$ has a quadratic zero on the diagonal, testimony to quadratic level repulsion.

$$= \delta(E_1 - E_2) \sum_{i=1}^{L} \delta(E_1 - x_i), \tag{42}$$

where in the last equality we enforced the limit $\sigma = 0$.

2. For large $\sigma$, the connected term in (34) simplifies by first rescaling $u_i \to 2u_i \sigma^2/b$ and then taking the large $L$ limit. This again simplifies the product over $i$ and $j$, the result is

$$\oint_0 \frac{du_1}{2\pi i} \oint_0 \frac{du_2}{2\pi i} \frac{1}{(u_1 - u_2)^2} \exp\left\{ \frac{ibt_1}{2}\left(u_1 + \frac{1}{u_1}\right) + \frac{ibt_2}{2}\left(u_2 + \frac{1}{u_2}\right) \right\}. \tag{43}$$

Expanding this out into powers of $u_1$ and $u_2$, by using (27), one obtains

$$\oint_0 \frac{du_1}{2\pi i} \oint_0 \frac{du_2}{2\pi i} \frac{1}{u_2^2} \sum_{n,m=0}^{\infty} \left(\frac{u_1}{u_2}\right)^n \left(\frac{u_1}{u_2}\right)^m \sum_{j,k=-\infty}^{+\infty} (iu_1)^j (iu_2)^k J_j(bt_1) J_k(bt_2). \tag{44}$$

The contour integral then picks up the terms with $j = -n - m - 1$, and $k = n + m + 1$ and we obtain

$$\langle Z(it_1) Z(it_2) \rangle \supset \sum_{l=0}^{\infty} (l+1)(-1)^{l+1} J_{l+1}(bt_1) J_{l+1}(bt_2). \tag{45}$$

Taking the Fourier transform before doing the sum, one recover the known wormhole contribution for the undeformed Gaussian matrix integral (with implicit Heavisides such

that the square roots remain real)

$$T(E_1, E_2) = \frac{1}{2\pi^2} \frac{b^2 - E_1 E_2}{(E_1 - E_2)^2} (b^2 - E_1^2)^{-1/2} (b^2 - E_2^2)^{-1/2}. \tag{46}$$

When the eigenvalues are close together this gives the universal answer

$$T(E_1, E_2) = \frac{1}{2\pi^2} \frac{1}{(E_1 - E_2)^2}. \tag{47}$$

Notice as consistency check also that this wormhole is order $L^0$.

We thus find that the wormhole in the completely averaged theory, ultimately becomes the diagonal delta functions (42) in the completely fixed theory. Of course, to find the delta functions one would have to include perturbative and nonperturbative corrections in $L$. One might wonder what the genus zero contribution to (42) is, i.e. the actual wormhole. Looking ahead, the results of section 2.3 suggest an educated guess

$$T(E_1, E_2) \overset{\text{guess}}{=} \frac{1}{2\pi^2} \sum_{i=1}^{L} \frac{4\sigma^2/L - (E_1 - x_i)(E_2 - x_i)}{(E_1 - E_2)^2} \times \tag{48}$$

$$\times (4\sigma^2/L - (E_1 - x_i)^2)^{-1/2} (4\sigma^2/L - (E_2 - x_i)^2)^{-1/2}. \tag{49}$$

Indeed as we discuss below, the genus zero spectrum becomes $L$ tight semicircles centered around each target eigenvalue $x_i$ and the above guess comes from treating each semi-circle independently. If two eigenvalues found themselves in the same semicircle, they would obviously still repel one another.

On a basic level one can notice, just from the plots, that $R(E_1, E_2)$ always has a quadratic zero on the diagonal $E_1 = E_2$; this means that if we look close enough, the quadratic Vandermonde repulsion is sill there. No matter how exotic one chooses an external potential for charged particles, at any finite $\sigma > 0$ there is always a distance scale where the electric repulsion between the particles wins, and one can forget the details of the external potential.

Another strong piece of evidence comes from the kernel (38). One can prove that, when the energies $E_1$ and $E_2$ are close enough together, this always reduces to the sine kernel

$$K(E_1, E_2) K(E_2, E_1) = \frac{\sin^2(\pi \rho(E_1)(E_1 - E_2))}{\pi^2(E_1 - E_2)^2}, \tag{50}$$

but now featuring the deformed spectrum [33, 36, 40]. Of course, the smaller $\sigma$, the closer together the eigenvalues must be for this formula to make sense. One should look on scales much smaller than the width $4\sigma/L^{1/2}$ of the tiny semicircles, but since the spectral density there is also huge, there should be a regime where one finds the universal wormhole answer (47). Here, the wormhole contribution should have only support on tiny regions centered around the target eigenvalues $x_i$; this logic suggests something like (49) makes sense.

This resonates well with earlier discussions about how gravitational systems can achieve factorization [11, 12, 60]. Factorization happens because the eigenvalue covariance $T(E_1, E_2)$ approaches a sum of delta functions on the diagonal, on a technical level. Indeed, the connected correlation is

$$\langle \rho(E_1) \rho(E_2) \rangle - \langle \rho(E_1) \rangle \langle \rho(E_2) \rangle = \delta(E_1 - E_2) \langle \rho(E_1) \rangle - T(E_1, E_2), \tag{51}$$

and this must vanish in non-random theories; this is clearly visible in Fig. 3.[3] In essence, the wormhole, which we define to include perturbative and nonperturbative corrections, becomes equal to the diagonal deltas when tuning towards $\sigma = 0$. See also section 5.

---

[3] Remember that $\langle \rho(E) \rangle$ itself also goes to delta functions in the non-averaged theory, see (25).

Most of the magic in this regard sits in the simplest observable, the spectrum. For any gravitational theory where the spectrum is sharply peaked, to good approximation, we know where all eigenvalues are; concordantly, the theory is almost non-random. This translates to the spectral covariance vanishing almost anywhere, except on the diagonal and close to these distinct points (where the spectrum is peaked). This is sufficient to obtain a factorizing theory since, according to (51), it means the connected correlation vanishes

$$\langle \rho(E_1)\rho(E_2) \rangle = \langle \rho(E_1) \rangle \langle \rho(E_2) \rangle \,. \tag{52}$$

## 2.3 Universal saddle for non-averaged gravity

The purpose of this subsection is working toward a gravitational interpretation for the theory at finite values of $\sigma$, we want to understand how one sees the peaks and valleys in figure 2 and figure 3 emerge. In particular, we want to highlight fundamental differences between the calculations in the completely averaged, and non-averaged cases.

In terms of the $u$ contour integrals, the difference is whether we do perturbation theory around poles at the origin $u = 0$ or immediately take residues at $u = x_i$. By resumming the perturbative expansion around the origin $u = 0$, one ultimately finds the contributions from all the poles; therefore, expanding around $\sigma = \infty$ and looking at the effects of high-order terms, is a sensible way to investigate the theory at finite $\sigma$. We do this in the double scaling gravity limit in section 4, and observe that spacetimes is shredded by many extra macroscopic boundaries [64], whenever the high-order terms become relevant.

Here we take a different path. There is a language where the transition from large to small $\sigma$ can be conveniently studied using saddle points. Nonperturbative effects in matrix integrals are best captured by another, dual matrix integral, sometimes known as the Efetov model [61, 62]. This has an interesting saddle point structure in which the plateau in the spectral form factor can be explained by a saddle known as the Andreev-Altshuler instanton [6, 62, 63, 80].

For this model one considers products of $K$ determinants in the matrix integral, then writes the determinants as Grassmann integrals, does the Gaussian integral over $H$; and uses a Hubbard-Stratonovich transformation with an auxiliary $K \times K$ Hermitian matrix $S$, to perform the Grassmann integrals. The result of these steps, for the undeformed Gaussian matrix integral (9) with $\sigma = \infty$, is [5, 62, 80, 81]

$$\left\langle \prod_{i=1}^{K} \det(E_i - H) \right\rangle = \int dS \, \det(E - iS)^L \exp\left( -\frac{2L}{b^2} \operatorname{Tr}(S^2) \right). \tag{53}$$

Upon double scaling, this becomes the Kontsevich matrix integral [6, 82, 83].

One might wonder whether, in the case of an external matrix $H_0$, one can derive similar expressions. Fortunately, it is not difficult to see that this is indeed the case; one just diagonalizes $H_0$ in (9) (by a unitary rotation of $H$), and then proceeds precisely as in the standard calculation. Omitting the detailed derivation, one finds

$$\left\langle \prod_{i=1}^{K} \det(E_i - H) \right\rangle = \int dS \, \prod_{j=1}^{L} \det\left(E - x_j a^2/4\sigma^2 - iS\right) \exp\left( -\frac{2L}{a^2} \operatorname{Tr}(S^2) \right). \tag{54}$$

As check, notice that when $\sigma = \infty$, we trivially recover (53). However, when $\sigma = 0$ (and thus $a = 0$), the theory localizes on $S = 0$. We may then simply evaluate the other pieces of the integrand on-shell, this results immediately in the correct non-averaged answer

$$\left\langle \prod_{i=1}^{K} \det(E_i - H) \right\rangle = \prod_{j=1}^{L} \prod_{i=1}^{K} (E_i - x_j) = \prod_{i=1}^{K} \det(E_i - H_0) \,. \tag{55}$$

Using the $u$ integral, the non-average answer corresponds with a complicated contour integral around the eigenvalues $x_i$. Using the Efetov model however, the non-averaged answers is obtained already from one saddle point at $S = 0$. Perturbative expansions of the $u$ integral are on similar footing as the genus expansion. High-order effects in the genus expansion encode the transition from ramp to plateau, but in the Efetov model this transition follows from a simple saddle point approximation [62]. The Efetov model is clearly the easiest language to capture nonperturbative effects in gravity, and the transition from large to small $\sigma$ is one of those effects.

Therefore let us investigate the saddle points of the Efetov model (54) is more detail, for simplicity let us consider only one determinant, where $S$ becomes one real parameter

$$\langle \det(E - H) \rangle = \int_{-\infty}^{+\infty} \mathrm{d}S \, \exp\left( -\frac{2L}{a^2}S^2 + \sum_{i=1}^{L} \ln\left( E - x_i \frac{a^2}{4\sigma^2} - \mathrm{i}S \right) \right). \tag{56}$$

More determinants do not result in more interesting structure. The saddle point equation is

$$\frac{4L}{a^2}\mathrm{i}S = \sum_{i=1}^{L} \frac{1}{E - x_i a^2/4\sigma^2 - \mathrm{i}S}. \tag{57}$$

The solutions determine the genus-zero resolvents of the matrix model [62, 80]. To see this, notice that

$$\partial_E \det(E - H) = R(E) \det(E - H), \tag{58}$$

with $R(E)$ the resolvent. Applying this identity to the Efetov model

$$\partial_E \langle \det(E - H) \rangle = \int_{-\infty}^{+\infty} \mathrm{d}S \sum_{i=1}^{L} \frac{1}{E - x_i a^2/4s^2 - \mathrm{i}S} \exp\left( -\frac{2L}{a^2}S^2 + \sum_{i=1}^{L} \ln\left( E - x_i \frac{a^2}{4\sigma^2} - \mathrm{i}S \right) \right), \tag{59}$$

and evaluating the integral to leading order on the saddle points (57); one sees that the leading order resolvent is indeed proportional to the Efetov saddle points

$$\langle R(E) \rangle = \frac{4L}{a^2}\mathrm{i}S, \tag{60}$$

with $S$ a solution to (57). Using the relation between the genus zero resolvent and the spectral curve of the matrix model [5], we find that the saddle point equations of the Efetov model, are equivalent to the spectral curve equations for our matrix model (1) with an external field. That spectral curve can be found in [47]

$$y = \frac{4E}{a^2} - \frac{4}{La^2}\sum_{i=1}^{L} \frac{1}{y - x_i/\sigma^2}, \tag{61}$$

which is indeed the same as (57), if one uses the relation between $R(E)/L = V'(E) - y$ [47].

For the undeformed theory $\sigma = \infty$ there are two solutions to (57)

$$\langle R(E) \rangle = \frac{2L}{b^2}E \mp \frac{2L}{b^2}(E^2 - b^2)^{1/2}, \quad \mathrm{i}S = \frac{E}{2} \mp \frac{1}{2}(E^2 - b^2)^{1/2}. \tag{62}$$

This is respectively the physical sheet of the resolvent for the Gaussian matrix integral, because there it decays as $L/E$ at large $E$, and the second sheet; obtained by going through the branch-cut. The spectrum is computed on the first sheet and gives the standard semicircle (29), from (62). Note that this saddles have real and imaginary parts in the allowed region.

Crucially however, for finite values of $\sigma$, the spectral curve has $L + 1$ solutions, corresponding to the different sheets of the resolvent for the matrix model with an external field [47].

When lowering $\sigma$, the other saddles will become competitive, and we believe they account for the heavy oscillations seen in the middle panel of Fig. 2 [62].[4]

Let us now discuss these other saddles both at large and small $\sigma$ in more detail. Solving (57) near $\sigma = \infty$ also reveals $L - 1$ extra non-physical saddle point solutions

$$\mathrm{i}S = E - \frac{q}{\sigma^2}, \quad \sum_{i=1}^{L} \frac{1}{q - x_i b^2/4} = 0, \tag{64}$$

on top of the standard saddles (62), which change only slightly near $\sigma = \infty$.

In the other extreme regime, close to $\sigma = 0$, the solutions to (57) (using (60)) can be described as follows. The solution on the physical sheet is given by

$$\langle R(E) \rangle = \sum_{i=1}^{L} \frac{L}{2\sigma^2}(E - x_i) - \frac{L}{2\sigma^2}((E - x_i)^2 - 4\sigma^2/L)^{1/2}, \tag{65}$$

again because it decays as $L/E$ at large $E$ and leads to a spectrum that consists of $L$ tight semicircles centered around each of the target eigenvalues $x_i$, as announced already around (49). To leading order in small $\sigma^2/L$ this is the saddle $S = 0$ announced below (54).

There are $L$ other solutions, where one of the relative signs in (65) is positive. These correspond to the second sheets of the same resolvent, having travelled through one of the $L$ tiny branchcuts of width $4\sigma/L^{1/2}$. These $L$ solutions behave for large $E$ as $(E - x_i)L/\sigma^2$. To leading order in small $\sigma^2/L$ these saddle points are $\mathrm{i}S = E - x_i$. In total this gives $L + 1$ solutions. We note that the leading order spectrum, coming from the saddle $S = 0$

$$\langle \rho(E) \rangle = \frac{L}{2\pi\sigma^2} \sum_{i=1}^{L} (4\sigma^2/L - (E - x_i)^2)^{1/2}, \tag{66}$$

already reproduces the delta functions (25) when $\sigma = 0$, the $\mathrm{i}S = E - x_i$ saddles seem to be redundant. We will indeed argue below that this is the case at the completely non-random point $\sigma = 0$.

In light of [80] note that $S = 0$ is the regime where the sigma model action vanishes and the universal features of a random matrix theory disappear, which makes sense if the saddle point $S = 0$ corresponds with the non-averaged theory. Notice also that products of determinants trivially factorize when the integral localizes on $S = 0$.

We want to know which of these saddles are actually on the integration contour for different values of $E$, and which saddles near $\sigma = 0$ flow towards which saddles near $\sigma = \infty$.[5]

Remarkably, the physical saddle $S \propto E - (E^2 - b^2)^{1/2}$ flows towards the $S = 0$ saddle. To appreciate this, consider some large energy $E$ that lies outside all cuts for any $\sigma$. At $\sigma = \infty$ and for energies in the forbidden region, only the saddle $S \propto E - (E^2 - b^2)^{1/2}$ lies on the integration contour [62]; we checked within a simple example in appendix A that this remains true near $\sigma = \infty$. Similarly we checked that near $\sigma = 0$, only the $S = 0$ saddle lies on the integration contour. Since $E$ by assumption never leaves or enters any cut, the saddle $S \propto E - (E^2 - b^2)^{1/2}$ near $\sigma = \infty$ must connect continuously to the $S = 0$ saddle near $\sigma = 0$. We therefore think of $S = 0$ also as the physical saddle.

---

[4] A detailed analysis of these effects requires defining the resolvent via

$$\langle R(E) \rangle = \lim_{M \to E} \partial_E \left\langle \frac{\det(E - H)}{\det(M - H)} \right\rangle. \tag{63}$$

Introducing inverse determinants replaces the bosonic matrix $S$ with a supermatrix, but the essence is unaffected: one needs to consider the other saddles for finite $\sigma$.

[5] We thank Steve Shenker for discussion on this.

The fun starts when considering energies $E$ that leave the spectral cut when changing $\sigma$. Say that for $\sigma < \sigma_c$, $E$ lies inside some cut, and that it lies outside all cuts for $\sigma > \sigma_c$. At this critical $\sigma$ we hit a (anti-)Stokes line and one saddle seizes to contribute, taking us from a real oscillating region in the determinant to an exponentially decaying one. The reverse phenomenon happens when $E$ enters a cut, another saddle starts contributing such that we obtain a real and oscillating determinant again. At small enough $\sigma$ the cuts are tiny, hence most $E$ will be outside the cut and only the $S = 0$ saddle contributes. We checked these statements explicitly for some simpler case where $\mathsf{H}_0$ has $L/2$ eigenvalues $z$ and $L/2$ eigenvalues $-z$, see appendix A.

Near $\sigma = 0$, as discussed above, the $L$ additional saddles besides $S = 0$ are given by $iS = E - x_i$. These other saddles (both close to $\sigma = 0$ and $\infty$) are problematic for the simple reason that they are purely imaginary and would give a contribution that is exponentially enhanced with energy $E$ squared. Based on this intuition and the simple example in appendix A, we therefore expect these saddles to not lie on the integration contour when $E$ is not inside any cut. When $E$ is inside some cut, we expect only the physical saddle (on the physical sheet) and the saddle where we went through the cut, in which $E$ lies, to contribute (recall below (65) that this is how you generate the non-physical saddles). We have checked this explicitly in simple examples like the one discussed in appendix A, leaving a more detailed check for the generic case to the future. [6]

Notably at $\sigma = 0$ there are no cuts. The takeaway message remains that the completely non-random point $\sigma = 0$ is described completely by one saddle point $S = 0$ in the Efetov model, which flows towards the physical saddle point $S \propto E - (E^2 - b^2)^{1/2}$, at the random matrix theory point $\sigma = \infty$.

## 2.4 Dispersion relation

The raison d'être of random matrices, is that the completely averaged description approximates many features of individual draws $\mathsf{H}_0$ of the system extraordinary well [62,74]; whilst being an exponentially simpler description. This shines through in the gravity dual: the bulk description of completely random systems $\sigma = \infty$ can be as simple as JT gravity [5], and the dual to eigenvalue repulsion are wormholes.

If one thing is certain it is that the gravity dual to some non-averaged system is much more complex, likely having some bulk action that is much more complicated that the JT gravity one. Nevertheless, one expects the wormhole to still be there and one question that has been raised [60] is how it can be seen in the non-averaged answer.

Since we have precise formulas for all correlation functions in the matrix model as function of $\sigma$, we can ask how the averaged contribution at large $\sigma$ is encoded in the small $\sigma$ behaviour.[7] This should help to understand what contributions one needs to include in order to go from, a non-factorizing theory to a factorizing one. The idea is to analytically continue $\sigma$ to the complex plane and use the following identity

$$\frac{1}{2\pi i} \oint_0 \frac{d\sigma}{\sigma} F(\sigma) + \frac{1}{2\pi i} \oint_\infty \frac{d\sigma}{\sigma} F(\sigma) + \frac{1}{2\pi i} \sum_{\sigma_i} \oint_{\sigma_i} \frac{d\sigma}{\sigma} F(\sigma) = 0, \tag{67}$$

where the $\sigma_i$ denote all non-analyticities of $F(\sigma)$ - this could include branchcuts. The contour integral around all non-analyticities obviously vanishes. The residues for the first two terms

---

[6] As an aside, we note that when two branch-points hit the real energy axis, the spectrum develops two $E^{1/3}$ edges. The physics near this edge is described by the Pearcey kernel or the Kontsevich matrix integral with a $S^4$ potential [45]; and corresponds with the $(3, 1)$ minimal gravity. When the eigenvalues of the external matrix are allowed to be complex, other edges can appear [45]; but here they do not.

[7] We thank Onkar Parrikar for discussion on this

give

$$F(0) = F(\infty) - \frac{1}{2\pi i} \sum_{\sigma_i} \oint_{\sigma_i} \frac{d\sigma}{\sigma} F(\sigma). \tag{68}$$

Here $F(\sigma)$ can be any correlation function, for example we could insert the wormhole $T(E_1, E_2)$ from (34). Then the statement is that the diagonal deltas $F(0)$ in some non-averaged theory (42), equals the wormhole $F(\infty)$ from the completely averaged theory (46), plus corrections from other poles that wash out upon averaging.

This is similar to the conclusions of [60], though with saddles instead of poles; a fortunate coincidence is the similar role played by the parameter $\sigma$ in that paper. There is also vaguely similar flavor to some of the discussion about poles as corresponding with geometries in [32].

To make things super concrete in a simple example, take the $L = 1$ case of (9), also known as the Gaussian integral

$$\mathcal{Z}(\sigma, h_0) = \int_{-\infty}^{+\infty} dh \, \exp\left( -\frac{2}{b^2} h^2 - \frac{1}{2\sigma^2} (h - h_0)^2 \right). \tag{69}$$

The exact partition function in this theory is

$$F(\sigma) = \left\langle e^{-\beta h} \right\rangle = \exp\left( \frac{\beta^2 \sigma^2 - 2\beta h_0}{2(1 + 4\sigma^2/b^2)} \right), \tag{70}$$

which indeed is (21) for $L = 1$. This interpolated between the averaged result

$$F(\infty) = \exp\left( \frac{b^2 \beta^2}{8} \right), \tag{71}$$

which smoothly decays with time; and the non-averaged result, which highly oscillates with time

$$F(0) = e^{-\beta h_0}. \tag{72}$$

We also see that $F(\sigma)$ has essential singularities at $\sigma_i = \pm i b/2$, which we need to account for in (68). The infinitely many contributions coming from this term will, when combined with the averaged $F(\infty)$, reproduce the the non-average result $F(0)$. To see this, we expand the exponential in $F(\sigma)$ (70) and explicitly compute the sum of the residues at $\sigma_i = \pm i b/2$

$$\frac{1}{2\pi i} \sum_{\sigma_i} \oint_{\sigma=\sigma_i} \frac{d\sigma}{\sigma} F(\sigma) = \sum_{\sigma_i} \sum_{k=1}^{\infty} \frac{b^{2k}}{8^k (k-1)! k!} \partial_\sigma^{k-1} \left( \left( \frac{\beta^2 \sigma^2 - 2\beta h_0}{\sigma + \sigma_i} \right)^k \frac{1}{\sigma} \right)_{\sigma=\sigma_i} \tag{73}$$
$$= F(\infty) - F(0).$$

One can check that this indeed agrees with the difference between the non-average and average answer, for instance by doing a Taylor expansion in $\beta$. Clearly both $F(0)$ and the contribution from the essential singularities are oscillating, and therefore non-self-averaging, for Lorentzian times.

There is a similar pattern for generic $L$, see for example (23) The only non-analyticities seem to appear when $\sigma_i = \pm i b/2$, in which case there is an essential singularity. There are an infinite number of contributions coming from these singularities which conspire with the average answer to give something factorizing.

The challenge, much like for the results of [60], is to find a gravity interpretation for the contributions from these other poles. This is far from obvious. We believe that a good place to start, would be taking $F(\sigma)$ to be the Efetov model (54). This model is the most natural language to study non-perturbative effects in gravity, it being basically an open universe field theory, and it might be manageable to give gravitational meaning to the poles there. Another

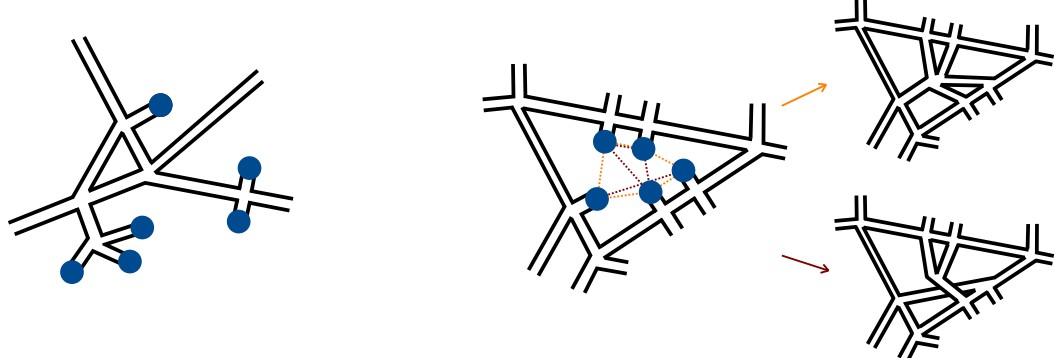

Figure 4: Ribbon graph of quartic matrix model with coupling $\text{Tr}(H_0 H)$ (left), the insertions of the external field $H_0$ (blue dots) are like leaves on a tree. For the gravity interpretation one must perform the unitary integral (right), giving a double sum over Wick contractions or permutations of which we show two examples (orange and red). The new vertices are weighed by Weingarten functions and traces of $H_0$, because of the Weingarten functions the orange contraction dominates for large $L$, as explained in detail in section 3.

avenue, would be to interpret $\sigma$ directly in gravity. Based on [60], perhaps it is similar to the $\Sigma_{LR}$ in the $G\Sigma$ formulation of SYK, since our $\sigma$ also tells us whether we are in a self-averaging or non-self-averaging region.

In the remainder of this work, we return to investigating the gravitational theory at finite $\sigma$ directly. We start with a discussion on ribbon graphs.

## 2.5 Ribbon graph intuition

Another way to get geometric intuition about matrix integrals, is to think about the ribbon graphs; or Feynman diagrams [84]. These are not that interesting in the pure Gaussian case, so let us temporarily consider a matrix integral with quartic interactions (148).

The external matrix coupling is weighed with $1/\sigma^2$ and is therefore expensive at large $\sigma$, concordantly there are barely insertions of the $H_0$ matrix in this regime. In terms of the ribbon graph, these insertions are vertices on which the ribbon graph ends, like the leaves of a tree; see Fig. 4. In the opposite regime of small $\sigma$, there are many such $H_0$ insertions [38].

This picture is however incomplete. As further discussed in section 5, these insertions of the matrix $H_0$ have no immediate gravitational interpretation; which is reserved for ribbon graphs that are built exclusively out of the field $H$. For that, we need to perform the integral over random unitaries (16) in (5); we diagonalize $H = U\Lambda U^\dagger$ and conveniently expand the exponential as

$$\int dU \exp\left(\frac{L}{\sigma^2} \text{Tr}\left(H_0 U \Lambda U^\dagger\right)\right) = \sum_{n=0}^{\infty} \frac{1}{n!} \left(\frac{L}{\sigma^2}\right)^n \int dU \, \text{Tr}\left(H_0 U \Lambda U^\dagger\right)^n. \tag{74}$$

These unitary integrals can be computed order per order using Weingarten functions [85,86]

$$\int dU \, \text{Tr}\left(H_0 U \Lambda U^\dagger\right)^n = \sum_{\sigma,\tau\in S_n} \text{Tr}_\sigma(H^n) \, \text{Tr}_\tau(H_0^n) \, \text{Wg}(\sigma\tau^{-1}, L), \tag{75}$$

which features a double sum over permutations in $S_n$. The notation for the traces should be intuitively clear

$$\text{Tr}_\sigma(H^n) = \prod_{\alpha_i} \text{Tr}\left(H^{l(\alpha_i)}\right), \tag{76}$$

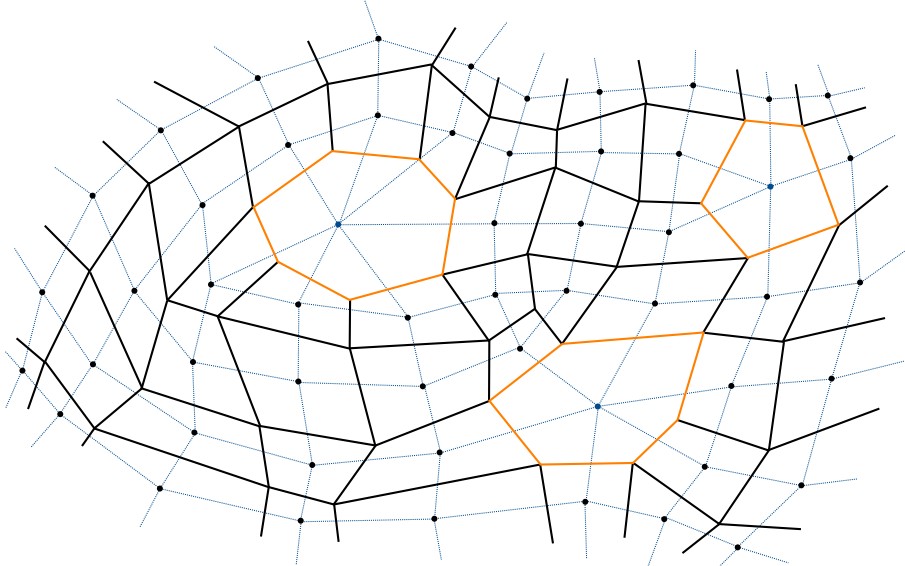

Figure 5: Discretized worldsheet for a quartic matrix integral with deformations, showing the quartic ribbon graph (blue) and the dual graph (black) which contains polygonic holes (orange) due to the deformations. The quartic interactions (black dots) are weighted by $\tau_4$ (as defined in B), the extra vertices (blue dots) from the deformations are weighed by traces of $\mathsf{H}_0$.

where $\alpha_i$ are the cycles of $\sigma$ and $l(\alpha_i)$ is the length of each cycle. The basic point is that in (75) one obtains all types of single-trace and multi-trace combinations of $H$. For example, the quartic matrix model acquires, when including the external field, all types of local vertices $\mathrm{Tr}(H^m)$; but also all multi-trace nonlocal vertices, like $(\mathrm{Tr}(H^p))^q$. The Feynman rules for these vertices, are set by combinations of $\mathrm{Tr}(\mathsf{H}_0^n)$. In conclusion, the external matrix determines the coupling constants of the deformed theory. This remains true for double-scaled gravitation theories as we discuss in section 3.

As discussed in section 3, single trace deformations of the matrix integral potential $\mathrm{Tr}(H^m)$ dominate at large $\sigma$. To see the related spacetimes, we must consider the dual graph. For the quartic matrix model this dual graph consists of squares, associated with the original interactions; and additional polygons representing the deformations $\mathrm{Tr}(H^m)$, each polygon is weighed by a coupling constant $\mathrm{Tr}(\mathsf{H}_0^m)$.

When we compute the partition function of the quartic theory with these deformations, the extra polygons are not interpreted as contributing to the Euler character; the topological expansion is one in powers of $\tau_4$, the quartic coupling constant for the undeformed theory, see equation (149). Therefore those polygons correspond with boundaries or holes of the spacetime [64], see Fig. 5.

Holes with order one valency become microscopic in the double-scaling limit, and correspond with local operators, or conical defects in gravity; holes with very high valency correspond with macroscopic boundaries in gravity. At large $\sigma$ the holes are isolated, meaning that the extra vertices are not adjacent, and holes with high valency are suppressed. For small $\sigma$ though, the extra holes can become adjacent; they can therefore condense and become macroscopic; also isolated large holes are no longer suppressed. This changes the spacetimes drastically, effectively tearing them up [64]. See section 4.

In the following sections we clarify how these statements translate to the double scaling limit, where the theory describes two dimensional dilaton gravity, as explained in section 1.

# 3 Deformed dilaton gravity

In the remainder of this work we study the effect of the external matrix $\mathsf{H}_0$ in the double scaling limit, and therefore in two dimensional gravity theories. This is a subtle endeavour, since $L$ is strictly infinite and naively, say, (22) becomes independent of $\sigma$. This means that, to find continuum limits with nontrivial dependence on $\mathsf{H}_0$, one must simultaneously carefully scale $\sigma$ too.

In this section we will describe one such scaling, relevant for large $\sigma$, where one can treat the external matrix in (5) as a perturbation, allowing us to investigate the effects of fixing $H$ ever so slightly.

The situation for small $\sigma$ is more mysterious. As discussed in section 2.3, the matrix integral develops many tiny cuts, making it is unclear what double scaling precisely means. We will make a compromise in section 5 and instead study a setup where only part of $\mathsf{H}_0$ is non-averaged, but most of this external matrix *is* random. In the resulting two-matrix model, we *can* find a continuum description for small $\sigma$.

We now start our investigation for large $\sigma$. For convenience, we give the matrix integral (5) again

$$\mathcal{Z}(\sigma, \mathsf{H}_0) = \int dH \, \exp\left( -L \operatorname{Tr} V(H) - \frac{L}{2\sigma^2} \operatorname{Tr}(H^2) + \frac{L}{\sigma^2} \operatorname{Tr}(\mathsf{H}_0 H) \right), \qquad (77)$$

and diagonalize $H = U \Lambda U^\dagger$. Since we are interested in trace class observables, the integral over Haar random unitaries is always the same one,

$$\int dU \, \exp\left( \frac{L}{\sigma^2} \operatorname{Tr}(\mathsf{H}_0 \, U \Lambda U^\dagger) \right). \qquad (78)$$

Previously we evaluated this integral exactly using the Harish-Chandra-Itzykson-Zuber formula [75, 76]. Throughout this section, however, we are interested in working close to infinite $\sigma$ and treat (78) perturbatively in $1/\sigma^2$. The exact formula (16) is not naturally suited for such an expansion.

To obtain an approximation at $\sigma \gg 1$, it is more efficient to instead use the trick

$$\left\langle \exp\left( \frac{L}{\sigma^2} \operatorname{Tr}(\mathsf{H}_0 \, U \Lambda U^\dagger) \right) \right\rangle = \exp\left( \sum_{n=1}^{\infty} \frac{1}{n!} \frac{L^n}{\sigma^{2n}} \left\langle \operatorname{Tr}(\mathsf{H}_0 \, U \Lambda U^\dagger)^n \right\rangle_{\text{conn}} \right), \qquad (79)$$

where the average denotes the Haar integral. This is an application of the general identity in statistics

$$\log\langle \exp(x) \rangle = \sum_{n=1}^{\infty} \frac{1}{n!} \langle x^n \rangle_{\text{conn}}. \qquad (80)$$

In physics we know this for example from the calculation of D-brane partition functions where $\langle x^n \rangle_{\text{conn}}$ would be the sum of all connected worldsheets with $n$ boundaries ending on the D-brane, and the $1/n!$ is because the boundaries are indistinguishable [5, 87]. The rewrite (79) is exact, if the sum converges. Whether it does or not, is an interesting question. In the approximation which we make here, we will see momentarily that it does converge, but this might no longer be the case when we transition to smaller $\sigma$, we comment on this in section 4 and 6.

As mentioned around (75), correlators of the Haar random ensemble are expressed in terms of Weingarten functions $\operatorname{Wg}(\sigma, L)$, which are known explicitly [85, 86]. We consider here the double scaling limit, where $L$ is sent to infinity. One may then use the large $L$ behavior of the Weingarten functions, to prove [12] that the leading large $L$ correlators of the

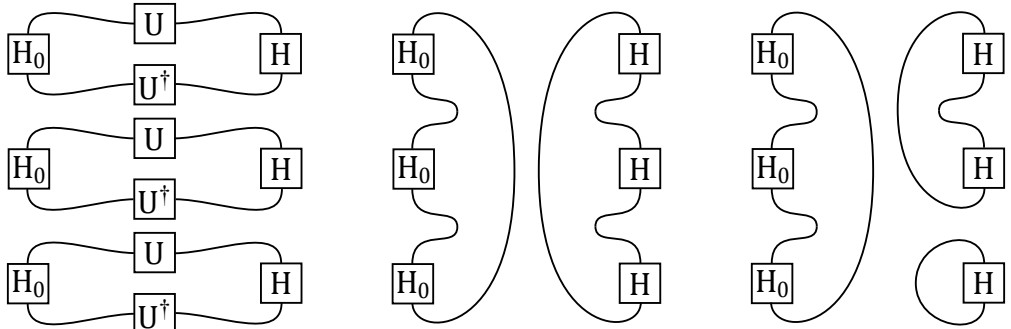

Figure 6: Illustration for integrals over unitaries. The wires represent summation over indices and integrating over random unitaries inserts a complete sets of wires. Weingarten functions $\mathrm{Wg}(\alpha\beta^{-1}, L)$ weight each bra-ket combination and are the inverse of the matrix of overlaps between wire states, which is the Gramm matrix $L^{\#(\alpha \cdot \beta^{-1})}$ [85] with $\#(\alpha)$ the number of cycles in the permutation $\alpha$. Dominant large $L$ configurations are diagonal contractions (middle) with identical bras and kets, whereas subleading configurations correspond with multi-trace operators (right).

Haar random ensemble, go to the Wick contractions of an ensemble of independent Gaussian complex variables with variance $L$.

Using this leading large $L$ behavior it is quite straightforward to compute each of the terms in (79), taking into account the discussion around (75)

$$\frac{L^n}{\sigma^{2n}} \Big\langle \mathrm{Tr}\big(\mathsf{H}_0 \, U \, \Lambda \, U^\dagger\big)^n \Big\rangle_{\mathrm{conn}} = \frac{1}{\sigma^{2n}} (n-1)! \, \mathrm{Tr}\big(\mathsf{H}_0^n\big) \, \mathrm{Tr}(H^n) - \frac{1}{L} \frac{1}{\sigma^{2n}} \times \tag{81}$$

$$\times \sum_{m=1}^{n} (m-1)!(n-m-1)! \, \mathrm{Tr}\big(\mathsf{H}_0^n\big) \, \mathrm{Tr}(H^m) \, \mathrm{Tr}\big(H^{n-m}\big) + \dots,$$

where the factorial counts the number of fully connected Wick contractions and we used $\mathrm{Tr}\, H^k = \mathrm{Tr}\, \Lambda^k$, see Fig. 6 for a graphical representation of these calculations. The subleading corrections come from the Weingarten functions $\mathrm{Wg}(\sigma, L)$ where $\sigma$ has multiple cycles, hence the emergence of multi-trace operators. In making this approximation, we have assumed that in the double scaling limit, all traces should *not* be interpreted as scaling with $L$. This is self-consistent concerning the $\mathrm{Tr}\big(\mathsf{H}_0^n\big)$; to implement the double scaling limit we will be urged below to scale these with the $n$th power of the spectral edge, and indeed with *no* extra overall $L$ associated with each trace.

The scaling of the $\mathrm{Tr}(H^n)$ is harder to establish, the procedure that we will use to analyze the double scaled scaled theory is insensitive to their scaling as long as multi-trace operators $\mathrm{Tr}(H^{n_1}) \, \mathrm{Tr}(H^{n_2})$ are negligible in the action. We assume here they are subleading at large $\sigma$, and comment on their potential significance for smaller $\sigma$ in section 6. in the remainder of this work we continue with (81).

Inserting (81) in (79) results in a deformed matrix integral

$$\mathcal{Z}(\sigma, \mathsf{H}_0) = \int \mathrm{d}H \, \exp\left( -L \, \mathrm{Tr}\, V(H) - \frac{L}{2\sigma^2} \, \mathrm{Tr}\big(H^2\big) + \sum_{n=1}^{\infty} \frac{1}{n} \frac{\mathrm{Tr}\big(\mathsf{H}_0^n\big)}{\sigma^{2n}} \, \mathrm{Tr}(H^n) \right)$$

$$= \int \mathrm{d}H \, \exp\left( -L \, \mathrm{Tr}\, V(H) - \frac{L}{2\sigma^2} \, \mathrm{Tr}\big(H^2\big) - \sum_{i=1}^{L} \mathrm{Tr}\log\big(\sigma^2/x_i - H\big) \right) \tag{82}$$

$$= \int \mathrm{d}H \, \frac{1}{\det(\sigma^2/\mathsf{H}_0 \otimes 1 - 1 \otimes H)} \, \exp\left( -L \, \mathrm{Tr}\, V(H) - \frac{L}{2\sigma^2} \, \mathrm{Tr}\big(H^2\big) \right), \tag{83}$$

up to irrelevant normalization factors. This can be viewed as a matrix integral with potential $V(H) + H^2/2\sigma^2$, with a stack of ghost-branes inserted, which are represented by the inverse determinant [5, 83].

Before we embark on double scaling of this matrix model, let us first study how the finite $L$ resolvent and spectral density are modified in the presence of inverse determinants.

### 3.1 Deformed resolvent and spectral density

The resolvent of a matrix model is defined as

$$R(E) = \text{Tr}\left(\frac{1}{E - H}\right),\tag{84}$$

and by taking the discontinuity across the real axis it gives the spectrum,

$$R(E + i\varepsilon) - R(E - i\varepsilon) = -2\pi i \rho(E),\tag{85}$$

whose normalization is determined by the total number of eigenvalues in the game

$$\int_{-\infty}^{+\infty} dE \, \rho(E) = L.\tag{86}$$

We are interested in the saddle point solution for $\rho(E)$; the genus zero spectral density. The saddle point equations for a matrix integral with potential $W(H)$ are [88]

$$LW'(E) = 2 \fint_{-E_0}^{+E_0} d\lambda \, \frac{\rho(\lambda)}{E - \lambda} = R(E + i\varepsilon) + R(E - i\varepsilon).\tag{87}$$

This should be satisfied only on the support of the saddle point solution for $\rho(E)$, which we will assume is a single connected region $[-E_0, E_0]$. We will consider the eigenvalues of $H_0$ to come in pairs $\pm x_i$ and so the full matrix potential (82) is even. This choice will clearly not affect the behavior near one of the edges, but it simplifies calculations because the spectrum becomes symmetric. The equation of motion (87), together with the constraint (86), are sufficient to solve for $R(E)$ and concordantly $\rho(E)$.

Indeed, imposing that $R(E)$ has a discontinuity only on the interval $[-E_0, E_0]$, that it decays as $L/E$ towards infinity, but has no poles elsewhere in the complex plane, one can invert (87) and find [5, 89]

$$R(E) = -\frac{L}{4\pi i} \oint_{\mathcal{C}} \frac{d\lambda}{\lambda - E} W'(\lambda) \sqrt{\frac{E^2 - E_0^2}{\lambda^2 - E_0^2}},\tag{88}$$

with $\mathcal{C}$ a contour around the spectral cut $[-E_0, E_0]$. Since this formula is linear in $W$, we can simply focus on the part of the potential coming from the inverse determinants separately (82)

$$\delta W'(\lambda) = -\frac{1}{L} \sum_{i=1}^{L} \frac{1}{\sigma^2/x_i - \lambda}.\tag{89}$$

The contribution to the resolvent from this deformation, denoted by $\delta R(E)$, is then [68]

$$\delta R(E) = -\frac{1}{4\pi i} \oint_{\mathcal{C}} \frac{d\lambda}{\lambda - E} \sum_{i=1}^{L} \frac{1}{\lambda - \sigma^2/x_i} \frac{(E^2 - E_0^2)^{1/2}}{(\lambda^2 - E_0^2)^{1/2}}.\tag{90}$$

We see that the integrand could potentially have poles on the spectral cut. We assume that $\sigma^4/x_i^2 > E_0^2$, such that all poles are outside the cut, see Fig. 7. This is identical to the convergence criterion of (79). Surprisingly, as discussed in section 4, this criterion $\sigma^4/x_i^2 > E_0^2$ is always satisfied within the approximation (82).

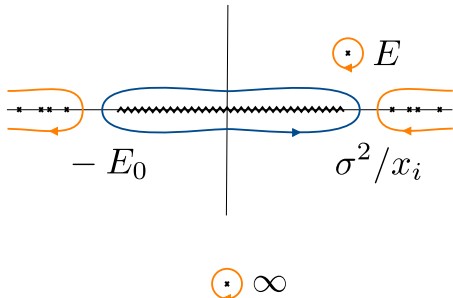

Figure 7: Contours and their deformation relevant for computing (90). In blue the initial contour $\mathcal{C}$ and in orange the deformed one.

We now deform the contour $\mathcal{C}$ around the poles at infinity, $E$ and $\sigma^2/x_i$. The residue at infinity vanishes because $\delta W'(\lambda)$ goes like $1/\lambda$ towards infinity. Combining the remaining residues, we obtain

$$\delta R(E) = \frac{1}{2}\sum_{i=1}^{L} \frac{1}{E - \sigma^2/x_i} - \frac{1}{2}\sum_{i=1}^{L} \frac{1}{E - \sigma^2/x_i} \frac{(E - E_0)^{1/2}}{(\sigma^2/x_i - E_0)^{1/2}} \frac{(E + E_0)^{1/2}}{(\sigma^2/x_i + E_0)^{1/2}}. \tag{91}$$

This expression is regular at the naive poles $\sigma^2/x_i$ because the residues vanish. Taking the discontinuity, one finds that the correction to the spectral density is given by

$$\delta\rho(E) = -\frac{1}{2\pi}\sum_{i=1}^{L} \frac{\mathrm{sgn}(x_i)}{\sigma^2/x_i - E} \frac{(E_0^2 - E^2)^{1/2}}{(\sigma^4/x_i^2 - E_0^2)^{1/2}}. \tag{92}$$

Here the square roots are to be interpreted as positive – we have extracted the explicit minus associated with the square root in the denominator of (88) and (91) when $\sigma^2/x_i$ lies to the left of the spectral cut [5,68]. Note that in our case of interest where the eigenvalues of the target Hamiltonian come in pairs $\pm x_i$, the spectral density remains even, and the spectral density has decreased. This is consistent with the deformation in the potential (82) [64]

$$\mathrm{Tr}\,\delta W(H) = \frac{1}{L}\sum_{i=1}^{L/2} \mathrm{Tr}\log\left(\sigma^4/x_i^2 - H^2\right), \tag{93}$$

which is also even and negative – obviously a shallower potential with the same number of eigenvalues filling it, gives a shallower and broader equilibrium sea of eigenvalues, therefore $E_0$ will have increased.

The full spectral density is thus given by

$$\rho(E) = \rho_V(E) - \frac{1}{2\pi}\sum_{i=1}^{L} \frac{\mathrm{sgn}(x_i)}{\sigma^2/x_i - E} \frac{(E_0^2 - E^2)^{1/2}}{(\sigma^4/x_i^2 - E_0^2)^{1/2}}, \tag{94}$$

with $\rho_V(E)$ the spectral density coming from $V(H) + H^2/2\sigma^2$ [68], which we compute explicitly for a quartic potential in appendix B. Note that at large $\sigma$ the second term goes away and we are back to the original matrix model defined by $V(H)$. The above spectral density still contains one free parameter $E_0$, which is fixed by the normalisation condition 86. For instance, if we take $V(H)$ to be quartic like in appendix B, $E_0$ needs to satisfy,

$$\frac{E_0^2}{4\tau}\left(1 - \frac{3}{4}\tau_4 E_0^2\right) - \frac{1}{2L}\sum_{i=1}^{L}\left(\left(1 - E_0^2\frac{x_i^2}{\sigma^4}\right)^{-1/2} - 1\right) = 1. \tag{95}$$

For $\sigma = \infty$, this reduces to the constraint (153) for a quartic matrix integral. Notice also the divergence when $\sigma^4$ hits $x^2 E_0^2$, with $x$ the larges eigenvalue of $\mathsf{H}_0$; we return to this in section 4.

## 3.2 Double scaling

We are now ready to double scale our matrix model and interpret $H_0$ in gravity. Luckily, the deformation of the potential is just a bunch of inverse determinants and the procedure for how to double scale this is well-known [54, 90]. We find it useful to review that here, especially since it will help us understand how to scale the parameters coming from the coupling to $H_0$, namely the eigenvalues of $H_0$ and $\sigma$.

The orienting discussion about branes will not be entirely rigorous, but we emphasize that our methods used to derive (107) *are* completely rigorous and consistent with results obtained using the string equation technology, which we discuss below in subsection 3.3 .

To start, it is useful to express the ghost-branes in terms of critical potentials [54].[8] This can be done using the following surprising identity [54, 83]

$$
\mathrm{Tr}\left(\frac{1}{y-H}\right) - \frac{L}{y} = \sum_{q=1}^{\infty} \frac{1}{y^{q+1}} \mathrm{Tr}(H^q)
$$
$$
= \sum_{k=1}^{\infty} (y+E_0)^{-k-1/2}(y-E_0)^{-1/2} \mathrm{Tr}\left((H+E_0)^{k-1/2}(H-E_0)^{1/2}\right)_+ , \quad (96)
$$

which holds for any choice of the constant $E_0$. The subscript $+$ means one should expand in powers of $1/H$ and keep only the terms with positive powers of $H$ in the resulting expansion. The $\mathrm{Tr}(\dots)_+$ in this expression are the critical potentials, or rather their derivative, $V'(H)$. To prove this one explicitly does the binomial expansions in $1/H$ and rearranges the resulting sums to collect all terms multiplying $\mathrm{Tr}(H^q)$, for fixed $q$. The remaining double sum for fixed $q$ equals $1/y^{q+1}$.

To double scale one then takes the constant $E_0$ to be the spectral edge, and considers energies close to this spectral edge

$$
H \to -E_0 + H, \quad y \to -E_0 + y, \quad (97)
$$

where $E_0$ is sent to $\infty$ whilst the new energies $H$ and $y$ remain finite. This double scaling then results in[9]

$$
\mathrm{Tr}\left(\frac{1}{y-H}\right) - \frac{L}{y} \stackrel{\mathrm{ds}}{=} \sum_{k=1}^{\infty} \mathcal{O}_{k-1} y^{-k-1/2}, \quad \mathcal{O}_k = \mathrm{Tr}\left(H^{k+1/2}\right)_+ . \quad (98)
$$

Applying the same logic to a stack of inverse determinants one obtains [83]

$$
\frac{1}{\det(Y \otimes 1 - 1 \otimes H)} \stackrel{\mathrm{ds}}{=} \exp\left(\sum_{k=0}^{\infty} \mathcal{O}_k \, t_k(Y)\right), \quad t_k(Y) = \frac{1}{k+1/2} \mathrm{Tr}(Y)^{-k-1/2}, \quad (99)
$$

up to an overall normalization constant that drops out in (82). Here, the operators $\mathcal{O}_k$ are known to correspond in the closed string worldsheet description with insertions of physical closed string operators, like the closed string tachyon vertices $\mathcal{T}_j$ of minimal strings [56, 57, 91–95].

The inverse determinant in our theory (82) has $Y = \sigma^2/H_0$. Since $H_0$ is the target Hamiltonian, we should scale it in precisely the same way as the random Hamiltonians and zoom in on target eigenvalues $x_i$ close to the spectral edge. The scaling of $Y$ in (97) is fixed by

---

[8] These are potentials that give rise to an $E^{m+1/2}$ spectral edge in the double scaling limit.

[9] This is the point in this derivation which is not rigorous, in the scaling (97) one secretly assumes that only order one eigenvalues of the shifted $H$ contribute. Whilst intuitively true, in formula (96) it is not clear that large eigenvalues of $H$ are suppressed in the term $(H - 2E_0)^{1/2}$.

demanding that the double scaled inverse determinant gives something nontrivial, together this demands we scale $\sigma$ and the $x_i$ like

$$\sigma \to E_0 + \sigma \,, \quad x_i \to -E_0 + x_i \,, \tag{100}$$

so that the coupling constants in (99) are given by

$$t_k(2\sigma + \mathsf{H}_0) = \frac{(-1)^{k+1/2}}{k+1/2} \operatorname{Tr}(2\sigma + \mathsf{H}_0)^{-k-1/2} \,. \tag{101}$$

If we now scale the theory with potential $V(H) + H^2/2\sigma^2$ to the critical point corresponding to the $(2,p)$ minimal string, the matrix integral (82) is a deformation around that by turning on the couplings $t_k(2\sigma + \mathsf{H}_0)$. In principle it is possible to translate these deformations (99) to linear deformations of the minimal string worldsheet action $I$

$$I_{\text{total}} = I - \sum_{j=0}^{\infty} \mathcal{T}_j \, \tau_j(Y) \,, \tag{102}$$

with these operators $\mathcal{T}_j$ implicitly integrated over the worldsheet. One could then rewrite these actions as dilaton gravity [56–58] and take the $p \to \infty$ limit to obtain a deformation of the JT gravity action (7).

In practice though, this is hard. The map between $t_k(Y)$ and $\tau_j(Y)$ is in general complicated, due to contact terms [91, 95]. Moreover because the sum runs over all $j$, we require not just the commonly studied tachyons,[10] but all physical closed string operators $\mathcal{T}_j$ – including those in the ground ring, and those with higher ghost numbers [93, 94]. These are much more mysterious, and seldom studied.

However, if we are interested in JT gravity, there is a much more efficient way of computing (7) that sidesteps the detour via the minimal string worldsheet formulation [65, 66, 68]. The idea is to scale the theory with potential $V(H) + H^2/2\sigma^2$ immediately to the critical point corresponding with JT gravity. We then simply solve the matrix integral (82) with the deformation due to the ghost-branes, meaning we calculate the genus zero spectral density – this completely specifies all genus amplitudes in matrix integrals [5, 48, 96]. Thanks to [49, 66] one can immediately map deformations around the JT gravity genus zero spectrum, to deformations of the dilaton gravity potential (7).

Let us return to the quartic matrix model. We will first tune the couplings so that the undeformed theory is tuned to criticality as $\tau_4 M_0^2 = g = 2/3(1 - 2\kappa/M_0)$. One could just double scale to the precise spectral edge $E_0$ of the deformed theory, however we want to understand to which degree $E_0$ changes with the deformation. Therefore, we should instead double scale to the spectral edge of the undeformed theory (149), here denoted $M_0$

$$E \to -M_0 + E \,, \quad E_0 \to M_0 - E_0 \,, \quad x_i \to -M_0 + x_i \,, \quad \sigma \to M_0 + \sigma \,. \tag{103}$$

In particular this means,

$$\frac{\sigma^2}{\mathsf{H}_0} \to -M_0 - (2\sigma + \mathsf{H}_0) \,, \tag{104}$$

with $x_i$ the eigenvalues of $\mathsf{H}_0$. Applying this to (92), one finds the spectral density

$$\rho(E) = \frac{2e^{\mathsf{S}_0}}{\pi} \left( \kappa(E - E_0)^{1/2} + \frac{2}{3}(E - E_0)^{3/2} \right) \tag{105}$$

---

[10] These correspond to the primary operators in the corresponding minimal model.

$$-\frac{1}{2\pi}\sum_{i=1}^{\infty}\frac{(E-E_0)^{1/2}}{(E-E_0)+(2\sigma+x_i+E_0)}\frac{1}{(2\sigma+x_i+E_0)^{1/2}}\,. \tag{106}$$

Notice that the deformation still vanishes for $\sigma=\infty$. We can expand this out as

$$\rho(E)=\rho_V(E-E_0)-\frac{1}{2\pi}\sum_{k=0}^{\infty}(E-E_0)^{k+1/2}\,\alpha_{k+1}(2\sigma+\mathsf{H}_0+E_0)\,, \tag{107}$$

where we have the coefficients

$$\alpha_k(2\sigma+\mathsf{H}_0+E_0)=(-1)^{k+1}\operatorname{Tr}(2\sigma+\mathsf{H}_0+E_0)^{-k-1/2}\,. \tag{108}$$

This result makes sense, because the whole point of inserting branes – here parameterized by $\sigma$ and $x_i$ – is that these can take us from any one minimal model to any other, or any deformation in between. Indeed the above is the most general expression for a double scaled spectral curve. Clearly the result (107) is valid for an arbitrary undeformed double scaled spectral curve, including that for JT gravity.

Note that the deformation parameters blow up when an eigenvalue of $\sigma^2/\mathsf{H}_0$ approaches the spectral cut. We connect this to the work of [64] in section 4.

Finally, one can determines $E_0$ by directly double scaling the constraint (86) or (95) and using (103) one obtains

$$0=2E_0(E_0+2\kappa)+e^{-\mathsf{S}_0}\sum_{i=1}^{\infty}\frac{1}{(2\sigma+x_i+E_0)^{1/2}}=2E_0(E_0+2\kappa)-e^{-\mathsf{S}_0}\alpha_0(2\sigma+\mathsf{H}_0+E_0)\,. \tag{109}$$

## 3.3 JT gravity

So far we have focused on the quartic matrix model, but from our discussion it is clear this works for any potential for the $(2,p)$ minimal string theories and by extension to $p=\infty$ also for JT gravity.

As an alternative to the above manipulations one could also employ the string equation technology which originates from the orthogonal polynomial approach to matrix models. This has the advantage of allowing a non-perturbative and numerical analysis [97], but the disadvantage of being more abstract. At any rate, it presents a useful check on the results of section 3.2.

The string equation is a differential equation for a function $u(x)$, which can be used to compute any correlation function [98, 99]. For the $(2,p)$ minimal string theories, the string equation takes the form,

$$x=\sum_k T_k R_k[u]\equiv\mathcal{F}[u]\,, \tag{110}$$

with $R_k[u]$ the Gelfand-Dickii functionals, which to leading order in the genus expansion go as $u^k$. The parameters $T_k$ are analogous to those we defined earlier in (99) and for the minimal string theories they take particular values, for instance see appendix B of [94] or [57]. We also defined $\mathcal{F}$ as the RHS of the string equation for convenience. It is not worthwhile for the present discussion to repeat or review the derivation of the string equation, but see [55] for a review. Let us denote by $\mathcal{F}_V$, the term in the RHS of the string equation coming from the potential without ghost-branes.

It is a simple application of the technology of [98, 100] to determine the effect of the ghost-branes. To leading order this gives the string equation,[11]

$$x=\mathcal{F}_V(u)-\frac{1}{2}e^{-\mathsf{S}_0}\alpha_0(2\sigma+\mathsf{H}_0+u)\,, \tag{111}$$

---

[11] The effect of branes scale with $e^{-\mathsf{S}_0}$ whereas the higher genus corrections start at $e^{-2\mathsf{S}_0}$.

with $\alpha_0$ given in (108). The equation for $E_0$ is obtained by setting $u = E_0$ and $x = 0$. This matches exactly with (109) when we use $\mathcal{F}(u) = u^2 + 2\kappa u$, the undeformed $(2,3)$ minimal string with non-zero cosmological constant. The density of states can then be computed using

$$\rho(E) = \frac{e^{S_0}}{2\pi} \int_{E_0}^{E} du \, \frac{\partial_u \mathcal{F}(u)}{\sqrt{E - u}}, \tag{112}$$

and we reproduce (107). This provides a check of the derivation in section 3.2. In the case of JT, we can use the results of [57] to find

$$\frac{\sqrt{E_0}}{2\pi} I_1(2\pi\sqrt{E_0}) - \frac{1}{2} e^{-S_0} \alpha_0(2\sigma + H_0 + E_0) = 0. \tag{113}$$

From these expressions we find that $E_0$ is negative, as anticipated around (93). In fact, it is subleading in $e^{S_0}$ and to leading order

$$E_0 = \alpha_0 e^{-S_0}, \tag{114}$$

where $\alpha_0$ is negative. To leading order, the spectral density for JT therefore takes the form

$$\rho(E) = \frac{e^{S_0}}{4\pi^2} \sinh\left(2\pi(E - E_0)^{1/2}\right) - \frac{1}{2\pi} \sum_{k=0}^{\infty} E^{k+1/2} \alpha_{k+1}(2\sigma + H_0). \tag{115}$$

## 3.4 Gravitational interpretation

With the preparatory work out of the way, we can discuss the gravitational interpretation of slightly fixing a member of the matrix integral ensemble. There are two ways to interpret our results; an open string picture which involves branes and the spacetime ending on it and a closed string picture, which captures our deformation as changing the dilaton gravity action. Both of them provide us with interesting insights as to what happens when one tries to collapse the matrix ensemble to one member.

### Open universes

We have learned that at large $\sigma$, the effect of the external matrix $H_0$ is just inserting a bunch of ghost-branes (83). In the double scaling limit there are infinitely many such branes, one for each eigenvalue of $H_0$. From a geometric point of view, this means that when we compute a certain observable, say the partition function $\langle Z(\beta) \rangle$, the sum over topologies includes spacetimes that not just have a large asymptotic boundary, but many other boundaries as well, since the spacetime can end on the ghost-branes. The boundary conditions on the brane side are of the FZZT type in the language of minimal string theory [5,83,101,102], and on the (classical) level of JT simply fixed energy boundary conditions [58].

For two point functions $\langle Z(\beta_1)Z(\beta_2) \rangle$, the presence of the branes gives rise to an explicit realisation of the idea of *broken cylinders* [103]; configurations which are disconnected and have some other boundary condition in the middle. The full sum over topologies is not yet factorizes, because $\sigma$ remains large, but it does indicate other contributions that might eventually take over and cause the two-point functions to factorize, see Fig. 8 for an illustration. Specifically, in this case one can see that the increasing number of brane boundaries *weakens* the geometric connection between the two asymptotic boundaries.

Notice also that our stack of ghost branes in the matrix potential is in spirit similar to the recently considered effective matrix model for dynamical end-of-the-world branes [68], those are D-branes with fixed mass Cardy state in open string parlance.

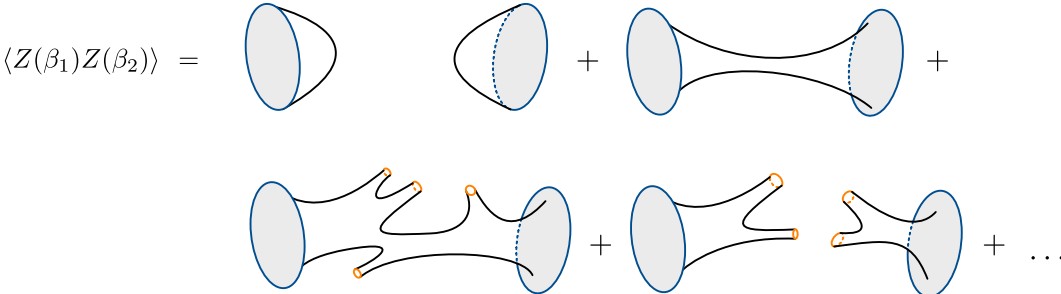

Figure 8: Contributions to $\langle Z(\beta_1)Z(\beta_2)\rangle$ in our gravity theory deformed by the external field $\mathsf{H}_0$. The blue boundaries are asymptotic boundaries, whereas the orange ones are boundaries come from the ghost branes and are labelled by the eigenvalues of $\mathsf{H}_0$ (not explicitly drawn here).

We have treated all ghost-branes as independent, and the boundaries associated to each eigenvalue exponentiate separately. This is an effect that happens at large $\sigma$, and arises because we took only the leading term in the unitary integral (81). The subleading corrections in (81) gives rise to double-trace terms for $\mathsf{H}_0$ and $H$, this means the RHS of (79) is no longer a single product over the eigenvalues of $\mathsf{H}_0$. At large $\sigma$, every eigenvalue of $\mathsf{H}_0$ can be seen as being associated with one ghost-brane, therefore the multi-trace terms that appear for smaller $\sigma$ can be thought of as interactions between the different ghost-branes. Indeed (16) is also not just a product over the eigenvalues of $\mathsf{H}_0$. We will discuss this more in section 6.

**Deformed dilaton potential**

The closed string interpretation is different. Using the results of [57, 65, 66], we can immediately map this deformation of the JT gravity spectral density (115) to a deformation of the dilaton gravity action (7). We find that the dilaton potential to order $e^{-\mathsf{S}_0}$ becomes

$$W(\Phi) = 2(\Phi + U(\Phi)), \quad U(\Phi) = -e^{-\mathsf{S}_0} e^{-2\pi\Phi} \sum_{k=0}^{\infty} \Phi^{2k} \alpha_k(2\sigma + \mathsf{H}_0) + \mathcal{O}(e^{-2\mathsf{S}_0}), \quad (116)$$

with $\alpha_k(2\sigma + \mathsf{H}_0)$ given in (108). Alternatively we can simply carry out the sum over $k$ and write

$$U(\Phi) = e^{-\mathsf{S}_0} \sum_{i=1}^{\infty} \frac{\sqrt{2\sigma + x_i}}{2\sigma + x_i + \Phi^2} e^{-2\pi\Phi} + \mathcal{O}(e^{-2\mathsf{S}_0}). \quad (117)$$

The insertion of the branes has thus been reinterpreted as small changes of the spacetime action. The most important take away from this section is there are perfectly sensible theories of dilaton gravity (116), which are less random than the simplest case of JT gravity.

Notice however that we have assumed here that $2\sigma + x_i > 0$. When $2\sigma$ becomes close to the largest (negative) eigenvalue we see that the corrections in $U$ become large and the dilaton potential seems to develop non-monotonicities. At that point however, one also needs to include higher genus corrections but not only to $U$ but also to $E_0$. A more thorough discussion of that is beyond the scope of the present discussion.

The fact that we have these two different ways of interpreting the effect of $\mathsf{H}_0$ in the bulk spacetime is a manifestation of an open-closed duality or as discussed in [60, 104] it is an explicit realisation where two bulk descriptions coexist.[12]

---

[12] The context is different than in [60], who discuss a path integral duality at small $\sigma$ whereas this duality is at large $\sigma$.

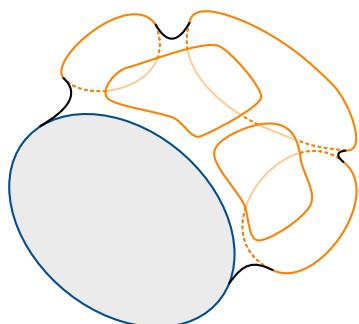

Figure 9: Disk that is torn apart because of the proliferation of macroscopic holes [64]. The blue boundary indicates an asymptotic boundary and the large orange holes are where the spacetime ends on ghost-branes. In reality these orange boundaries are much larger and the spacetime just consists of thin strips.

The large $\sigma$ regime showed what it means in gravity to slightly fix a Hamiltonian in the boundary matrix ensemble. However this is still only an asymptotic region of $\sigma$ space, and our main interest is in small $\sigma$. We next consider what happens when we back away from asymptotically large $\sigma$.

## 4 Tearing spacetime

When lowering $\sigma$, the coupling constants $\alpha_k$ in (108) blow up, when one of the eigenvalues of $\sigma^2/H_0$ approaches the spectral edge $E_0$, resulting in a proliferation of operator insertions. More importantly, operators $\mathcal{O}_k$ in (99) with large $k$ are strongly suppressed for large $\sigma$, when all eigenvalues are far from the cut, but this suppression stops when one of the eigenvalues approaches the edge, and operators with large $k$ dominate. Another way of seeing this proliferation is by noticing that the series (79) in (82) is no longer convergent for small $\sigma$, meaning that $\text{Tr}(H^k)$ operators for large $k$ dominate. These correspond to macroscopic holes in the spacetime, unlike local operators which have small $k$. The result is a spacetime with many large holes (see Fig. 9) which therefore appears to be torn apart.

Remarkably, this tearing phenomenon has been discovered by Kazakov [64], in the quartic matrix integral with avant la lettre ghost-brane-insertions. He studies the potential (149) with the deformation (99), but restricted to one eigenvalue pair. We have $L/2$ eigenvalue pairs, restricting to the case with an even spectrum

$$W(H) = \frac{1}{2\tau}H^2 - \frac{\tau_4}{4\tau}H^4 + \frac{1}{L}\sum_{i=1}^{L/2}\text{Tr}\log\left(\sigma^4/x_i^2 - H^2\right).$$ (118)

Here the parameter $\tau$ is understood to be fixed once and for all to the value

$$\tau = \frac{M_0^2}{8}.$$ (119)

This is the combination of (153) and (156), with $M_0$ the undeformed spectral edge (103). This was also used in (109).

Let $x$ be the absolute value of the largest negative eigenvalue of $H_0$.[13] Based on the above discussion one might expect a tearing phase transition when $\sigma^2/x = E_0$; however, a more

---

[13] As before, we will be scaling towards the left edge.

careful analysis shows that this transition happens when an eigenvalue of $\sigma^2/\mathsf{H}_0$ passes the edge of the *undeformed* spectrum

$$\frac{\sigma^2}{x} < M_0 . \tag{120}$$

Let us explain this in a bit more detail, and discuss the double-scaled continuum theory at both sides of the transition.

The point is that, when $\sigma$ becomes too small, the critical coupling for the theory, where one obtains the continuum $(2,3)$ minimal gravity theory, is no longer $\tau_4 = 1/12\tau$ and one needs to scale towards a different coupling constant to find a continuum theory [55]. To see this, consider the constraint equation (95)

$$\frac{g}{4\tau\tau_4}\left(1 - \frac{3}{4}g\right) - \frac{\gamma}{2}\sum_{i=1}^{1/\gamma}\left(\left(1 - \frac{g x_i^2}{\tau_4\sigma^4}\right)^{-1/2} - 1\right) = 1 , \tag{121}$$

where we introduced $g = \tau_4 E_0^2$ and following Kazakov introduced $\gamma = 1/L$. Furthermore, consider the derivative of the constraint equation with respect to $g$

$$\left(1 - \frac{3}{2}g\right) - \frac{\gamma\tau}{\sigma^4}\sum_{i=1}^{1/\gamma} x_i^2\left(1 - \frac{g x_i^2}{\tau_4\sigma^4}\right)^{-3/2} = 0 . \tag{122}$$

Naively taking $\gamma = 0$, one recovers the critical couplings $g = 2/3$ and $\tau_4 = 1/12\tau$. This second equation (122), tunes the coupling such that one obtains a $E^{3/2}$ spectral edge, and is analogous to demanding that the first term vanishes in (154), as is explained in the refreshingly didactic review [55].

However, as Kazakov explained, the limit $\gamma = 0$ is treacherous [64]. To see this, one can solve these equations perturbatively in $\gamma$, the first subleading correction gives

$$g = \frac{2}{3} - \gamma \frac{1}{12}\frac{M_0^2}{\sigma^4}\sum_{i=1}^{\infty}\left(1 - \frac{M_0^2 x_i^2}{\sigma^4}\right)^{-3/2} , \tag{123}$$

with a structurally similar expression for $\tau_4$. This expansion is regular when $\sigma^2/x > M_0$, but it becomes singular, and hence nonphysical, once this largest eigenvalue enters the undeformed cut $\sigma^2/x < M_0$ as follows from the negative fractional power.

This means that for $\sigma^2/x < M_0$ the critical couplings at $\gamma = 0$ are not $g = 2/3$ and $\tau_4 = 1/12\tau$, one should instead expand around different values to obtain an expansion with real couplings. The trick is to expand the couplings close to the singular point in (121) and (122), where one obtains the leading answer

$$E_0^2 = \frac{g}{\tau_4} = \frac{\sigma^4}{x^2} - \gamma^{2/3}\frac{1}{4}\left(1 - \frac{\sigma^4}{M_0^2 x^2}\right)^{-2/3} . \tag{124}$$

The power of $\gamma^{2/3}$ for the correction is an ansatz which implies the second term in (122) is order $\gamma^0$ and therefore competitive with the first term, and similarly in (121). The solution for the coupling itself is more messy, but has a similar structure $\tau_4 = a + \gamma^{2/3}b$ where $a$ and $b$ functions of $\sigma, x$ and $M_0$ that are real as long as $\sigma^2/x < M_0$. See Fig. 10 for an numerical solution to the constraints (95) and (122) as a function of $\sigma$. The transition is clearly visible there. For future purposes we note that $\partial_\sigma a \neq 0$.

Now for Kazakov's surprise. Using intuition from the discrete ribbon graphs, one deduces that the average circumference $\ell$ of the holes associated with the deformation (see Fig. 5) is proportional to [64]

$$\ell \propto -\frac{\sigma\partial_\sigma\tau_4}{\gamma\partial_\gamma\tau_4} \propto \gamma^{-2/3} . \tag{125}$$

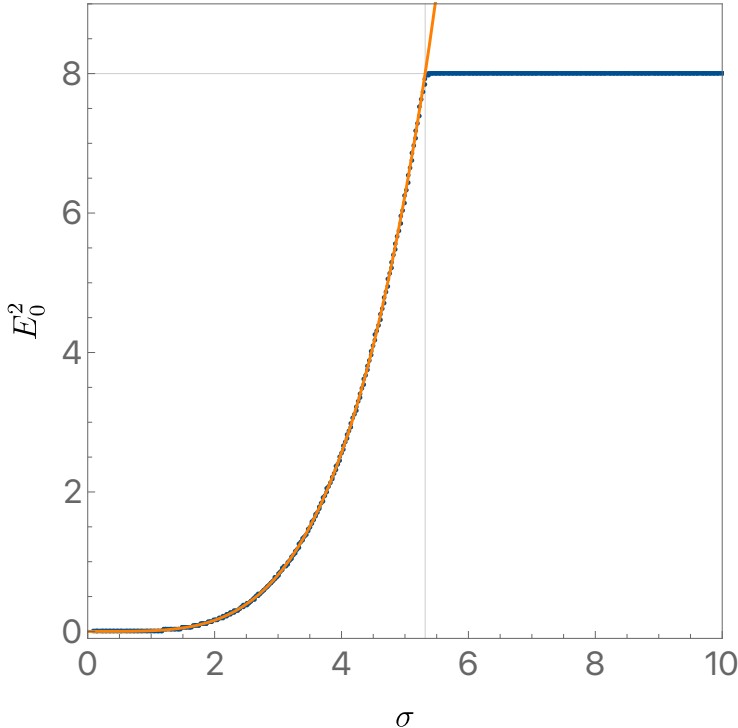

Figure 10: Tearing transition as a function of $\sigma$. Blue dots are numerical solutions to the (95) and (122) for $x = 10$, $\gamma = 10^{-4}$ and $M_0^2 = 8$. At large $\sigma$ we see the usual solution for the quartic matrix model and $E_0^2 \approx 8$ (horizontal grey line). As $\sigma$ is lowered and crosses $\sigma^2 = xM_0 = 20\sqrt{2}$, there is a first order phase transition and $E_0^2$ goes like $\sigma^4/x^2$, which is indicated with the solid orange curve. Note that at finite $\gamma$ the transition is smoothed out, but becomes sharp as $\gamma \to 0$.

In the limit $\gamma = 0$ these become macroscopic or even asymptotic boundaries; this is the *tearing phase* where the smooth spacetime is shredded by these large holes. In the phase before the tearing transition $\sigma^2/x < M_0$, the holes remain relatively small [64].

Notice that (124) implies that $\sigma^2/x_i < E_0$ everywhere. The eigenvalues $\sigma^2/x_i$ therefore never actually enter the spectral cut, the edge moves along; this validates using (107) for all values of $\sigma$, it protects the coupling constants from becoming imaginary and hence nonphysical.

From this analysis, we see that the undeformed potential does not affect the tearing transition. The non-trivial feature of this phase, the fractional power of $\gamma$ and $\partial_\sigma a \neq 0$, just comes from the addition of the brane terms. Consequently, the tearing phase is also present if we take the undeformed potential to be the one corresponding to JT gravity.

One might wonder what happens after $\sigma^2$ has crossed $M_0 x$. First, notice that the large boundaries that occur are labelled only by $x$. The other $x_i$ boundaries are still small, but when $\sigma^2$ becomes smaller, also those can become large. As a result, the surface becomes more and more torn. Second, when $\sigma$ becomes sufficiently small, the approximations we made to find (82) breaks down. For instance, multi-trace terms will become important, see section 6. Furthermore, we have not considered non-perturbative effects and the insertion of the Gaussian around $\mathsf{H}_0$ does not introduce any pathologies and so we expect that non-perturbatively this transition might be resolved. For instance, to check this one can compute the relevant (ghost) brane correlators in the Airy case [105], which are presumably smooth functions of the ghost brane energies $\sigma^2/x_i$.

# 5 Towards non-averaged dilaton gravity

We have just seen that the gravitational interpretation becomes more complicated when backing away from asymptotically large $\sigma$. Surprisingly though, in the other extreme of small $\sigma$, one can still find a gravitational interpretation, but this requires taking a slightly different route.

We propose for small $\sigma$ to modify (1) by integrating over $H_0$ and insert a small number of spectral densities for $H_0$, which fixes several eigenvalues of $H_0$ but leaves most of the Hamiltonian random. The matrix model we shall consider is

$$\mathcal{Z}(\sigma, \kappa_1 \dots \kappa_n) = \int dH \int dH_0 \, \text{Tr} \, \delta(H_0 - \kappa_1) \dots \text{Tr} \, \delta(H_0 - \kappa_n) \times$$
$$\times \exp\left(-L \, \text{Tr} \, V(H) - \frac{L}{2\sigma^2} \text{Tr}(H_0 - H)^2\right), \quad (126)$$

where $n \ll L$. At finite $\sigma$, we are dealing with a certain two-matrix model. At small $\sigma$, the Gaussian centered around $H_0$ becomes a delta function, and the $H_0$ integral collapses, giving an ordinary matrix integral with a bunch of densities inserted. This is the merit of integrating over $H_0$, as most of the eigenvalues of $H$ remain random, even at small $\sigma$ and so a more feasible direction to discuss a gravitational interpretation opens up.

## 5.1 Local factorization

Actually, to further motivate studying (126), we note that partial fixing is already enough to understand questions such as factorisation [5, 8, 10–12, 90]. As we explain now, this is because in energy space it results in what one could call local factorization.

Consider the spectral correlation for $n = 1$, to which we restrict during most of this section

$$\langle \rho(E_1)\rho(E_2)\rangle_\kappa = \frac{1}{\mathcal{Z}(\sigma, \kappa)} \int dH \, \text{Tr} \, \delta(H - E_1) \, \text{Tr} \, \delta(H - E_2) \int dH_0 \, \text{Tr} \, \delta(H_0 - \kappa) \times$$
$$\times \exp\left(-L \, \text{Tr} \, V(H) - \frac{L}{2\sigma^2} \text{Tr}(H_0 - H)^2\right). \quad (127)$$

One of the eigenvalues of $H$ is gradually fixed to $\kappa$; to appreciate this, notice that for small $\sigma$ we obtain a delta function $\delta(H_0 - H)$. By permuting the eigenvalues of $H$ one finds that indeed one eigenvalue has been fixed

$$\mathcal{Z}(\sigma, \kappa) = L \int_{-\infty}^{+\infty} \prod_{i=1}^{L} d\lambda_i \exp\left(-L \sum_{i=1}^{L} V(\lambda_i)\right) \Delta(\lambda)^2 \, \delta(\lambda_1 - \kappa). \quad (128)$$

The same thing happens in all correlators, and it carries over immediately to generic $n$.

The connected part of (127) is

$$\langle \rho(E_1)\rho(E_2)\rangle_{\kappa \text{ conn}} = \langle \rho(E_1)\rho(E_2)\rangle_\kappa - \langle \rho(E_1)\rangle_\kappa \langle \rho(E_2)\rangle_\kappa$$
$$= -T_\kappa(E_1, E_2) + \delta(E_1 - E_2)\langle \rho(E_1)\rangle_\kappa, \quad (129)$$

where one computes $\langle \rho(E_1)\rangle_\kappa$ analogously to how the two-point function is computed, but now with only the one insertion of $\text{Tr} \, \delta(H - E)$. Following the logic of section 2, we are interested in calculating $T_\kappa(E_1, E_2)$. Define thereto the sine-kernel [106], which features the undeformed (associated to a matrix integral with potential $V(E)$) spectral density $\rho(E)$

$$S(E_1, E_2) = \frac{\sin(\pi \rho(E_1)(E_1 - E_2))}{\pi(E_1 - E_2)}. \quad (130)$$

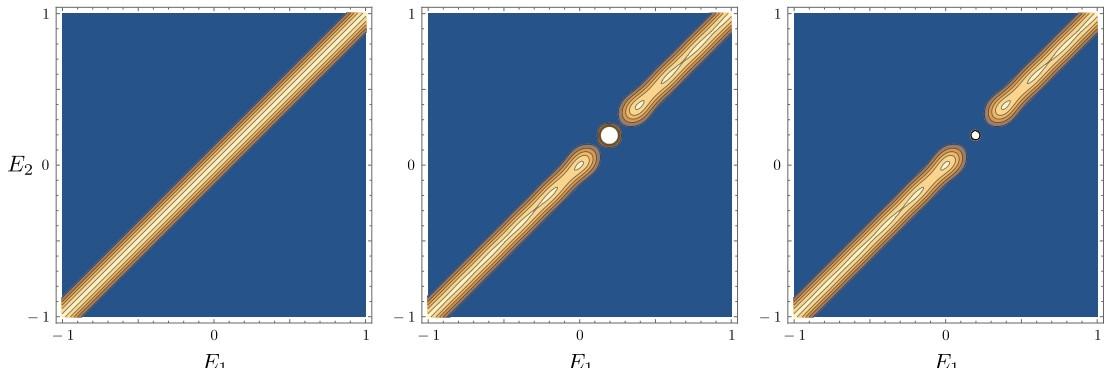

Figure 11: The spectral correlation $T(E_1, E_2)$ for the theory where we gradually fix one eigenvalue (127). For $\sigma = \infty$ (left) this is the characteristic sine-kernel of random matrix theory (131). For $\sigma = 0$ (right) the spectral correlation is completely destroyed close to the fixed eigenvalue, modulo a delta spike (132). For finite but small $\sigma$ (middle) the leading effect is a Gaussian smearing of this delta spike (143). Here $\kappa = 0.2$.

In the case $n = 0$ random matrix universality implies that one can approximate the covariance $T(E_1, E_2)$ for the completely random theory as [106]

$$T(E_1, E_2) = S(E_1, E_2)^2. \tag{131}$$

Using formula (67) of [11] one finds a similarly universal expression for the theory with one eigenvalue fixed

$$T_\kappa(E_1, E_2) = (S(E_1, E_2) - S(E_1, \kappa)S(E_2, \kappa)/S(\kappa, \kappa))^2 + \delta(E_1 - \kappa)\delta(E_2 - \kappa), \tag{132}$$

see Fig. 11. When $E_1$ and $E_2$ approach $\kappa$, the smooth part of the covariance vanishes

$$T_\kappa(E_1, E_2) = \delta(E_1 - \kappa)\delta(E_2 - \kappa) + \frac{\pi^4 \rho(\kappa)^6}{9}(E_1 - \kappa)^2(E_2 - \kappa)^2 + \dots . \tag{133}$$

This shows that fixing one eigenvalue $\kappa$ already destroys all spectral correlation for energies close to $\kappa$. In fact, it is clear that locally near this eigenvalue $\langle\rho(E_1)\rho(E_2)\rangle_\kappa$ already factorizes and gives rise to an interesting constraint between geometries.

Geometrically, (132) features 3 topologies; suppressing the genus expansion and the corresponding nonperturbative corrections, see Fig 12.[14] There is the wormhole connecting the two boundaries $\rho(E_1)$ and $\rho(E_2)$, the three holed sphere connecting $\rho(E_1)$ and $\rho(E_2)$ to $\rho(\kappa) = S(\kappa, \kappa)$, a product of two wormholes connecting $\rho(E_1)$ to $\rho(\kappa)$ and the second wormhole connecting $\rho(E_2)$ to a second copy of $\rho(\kappa)$. This last term originates from subtracting the disconnected terms in (129).

If we are close to $\kappa$, (133) tells us that $T_\kappa$ is small and we that the three aforementioned geometries need to satisfy the constraint as sketched in Fig. 12. When fixing multiple consecutive eigenvalues $\kappa_i$, the region where the corresponding $T_{\kappa_1 \dots \kappa_n}$ is small, grows and leads to more intricate relations between different geometries. We emphasize that the nonperturbative corrections are crucial for recovering these sine-kernel formulas, and the resulting factorization. Classical geometries will not explain Fig. 12.

---

[14] Here we use the dictionary between double scaled matrix models and minimal string theory to relate a insertion of the spectral density to a geometry with fixed energy boundary conditions [5]. In the JT limit such boundary conditions were studied also classically in [58].

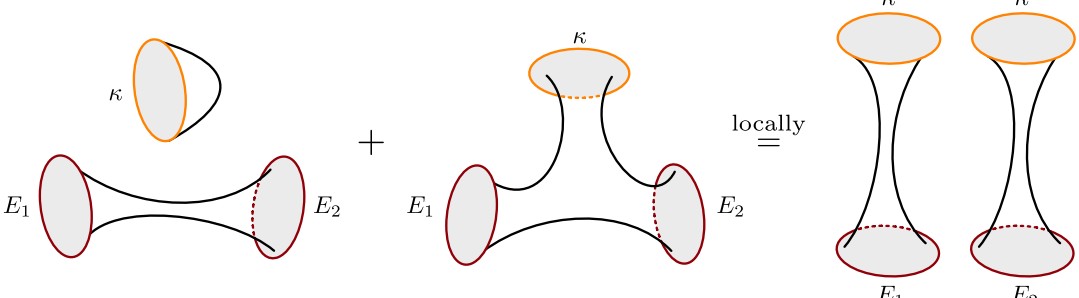

Figure 12: The geometries contributing to the spectral correlation (132), the genus expansion is suppressed for presentation purposes. Fixed energy boundaries are red and eigenbrane boundaries are orange. The first two terms should be normalized by one eigenbrane disk (labelled by $\kappa$), the third term is normalized by two such disks. Here we again emphasize that the relation between these geometries holds locally.

In connection to the dispersion relation in section 2.4, and [60] we notice that the self-averaging wormholes contribution is always there. The other two geometries strongly depend on $\kappa$ and are non-self-averaging. The third geometry represents the completely factorized diagonal contribution, it equals the sum of the wormhole and some other non-self-averaging geometry [11, 12].

Let us emphasize the main point. Suppose one considers JT gravity with one eigenvalue fixed at position $\kappa$, and computes the two point function of asymptotic boundaries with fixed energy boundary conditions. Then when considering boundary energies close to $\kappa$ one finds that this amplitude essentially factorizes. This makes the theory with only several fixed eigenvalues (126) worth understanding.

## 5.2 Gravitational interpretation

Let us now return to studying finite $\sigma$ (126). As mentioned before, this matrix model can be interpreted geometrically as having $n$ background boundaries labelled by $\kappa_i$. In [11, 12] they were dubbed eigenbranes as they represent fixed energy boundary conditions. When inserting probe boundaries $\rho(E_i)$, labelled by energies $E_i$, in order to compute various correlation functions, we sum over over all spacetimes that are consistent with the boundary conditions. We already saw an example of this in the previous subsection.

Unfortunately, these $n$ background boundaries have no immediate gravitational interpretation, because the auxiliary random matrix $H_0$ has no direct gravitational interpretation. We need to integrate $H_0$ out in order to make contact with gravity. Luckily, using the formulas from section 2, we can easily perform these $H_0$ integrals and obtain the appropriate insertion in the matrix integral for $H$.

The goal is understanding the gravitational dual of that insertion and how the eigenbrane picture is modified for nonzero $\sigma$. This gives us a better handle on the full parameter space of our theory (1). To simplify the analysis and discussion, let us focus on fixing just one eigenvalue. The insertion in the $H$ matrix integral is thus

$$\rho_{H_0}(\kappa) = \int dH_0 \, \mathrm{Tr} \, \delta(H_0 - \kappa) \exp\left(\frac{L}{2\sigma^2} \mathrm{Tr}(H_0 - H)^2\right). \tag{134}$$

The idea is to write this in terms of operators with known gravitational duals. For the purposes of this section we introduce a coupling $g^2 = L/\sigma^2$ which remains finite throughout; this is a different scaling of $\sigma$ than in section 3.

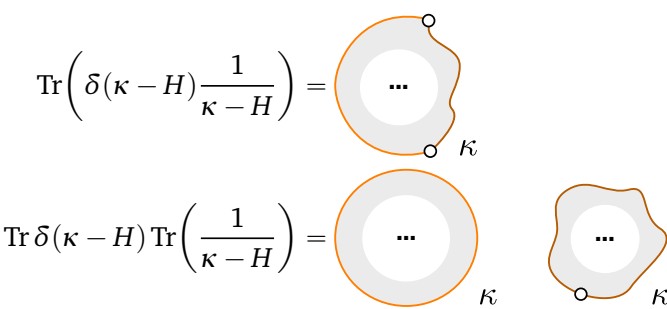

$$\mathrm{Tr}\left(\delta(\kappa - H)\frac{1}{\kappa - H}\right) =$$

$$\mathrm{Tr}\,\delta(\kappa - H)\,\mathrm{Tr}\left(\frac{1}{\kappa - H}\right) =$$

Figure 13: Contact terms correspond in gravity with different boundary conditions on different segments (left).The gravitational translation of the double trace operator (right) is inserting one boundary with fixed energy boundary state, and a marked FZZT boundary or resolvent [5, 107]. We distinguish these close related boundary states by picturing the FZZT boundary segments using dark orange wiggly curves in contrast to the eigenbrane boundary in orange.

By slightly modifying the derivation of (22) one can exactly perform this Gaussian integral and find [36]

$$
\rho_{\mathsf{H}_0}(\kappa) = \int_{-\infty}^{+\infty} \frac{\mathrm{d}t}{2\pi} \sum_{k=1}^{L} \exp\left(-\frac{1}{2g^2}t^2 + \mathrm{i}t(\lambda_k - \kappa)\right) \prod_{i \neq k}^{L} \frac{\lambda_k + \mathrm{i}t/g^2 - \lambda_i}{\lambda_k - \lambda_i}
$$
$$
= g^2 \int_{-\infty}^{+\infty} \frac{\mathrm{d}s}{2\pi} \oint_H \frac{\mathrm{d}u}{2\pi\mathrm{i}} \exp\left(-\frac{g^2}{2}(s^2 + (u - \kappa)^2)\right) \frac{1}{\kappa - \mathrm{i}s - u} \frac{\det(\kappa - \mathrm{i}s - H)}{\det(u - H)}, \quad (135)
$$

where the contour integral around the eigenvalues $\lambda_k$ of $H$, and we defined $s = \mathrm{i}(u - \kappa) - t/g^2$, which is actually the same variable as appearing in the Kontsevich integral. It is important to notice that here we are thinking about $H$ as a matrix that we still need to integrate over, so at this point it has a bunch of discrete eigenvalues and the above manipulations make sense.

The contour for the $u$ integral can be deformed so as to take the discontinuity, on the real axis, of the ratio of determinants (and not of the pole at $\kappa - \mathrm{i}s$). This is non-zero as a result of the Sokhotski-Plemelj theorem.

To find these discontinuities is difficult, but luckily, at large $g$ (small $\sigma$), the Gaussians in (135) are sharply peaked around $s = 0$ and $u = \kappa$. This invites us to Taylor expand the determinants as

$$
\frac{\det(\kappa - \mathrm{i}s - H)}{\det(u - H)} = \det\left(1 + \frac{\kappa - \mathrm{i}s - u}{u - H}\right) = 1 + \sum_{n=1}^{\infty} \frac{1}{n!}(\kappa - \mathrm{i}s - u) \sum_{i_1 \neq \cdots \neq i_n}^{L} \frac{1}{u - \lambda_{i_1}} \cdots \frac{1}{u - \lambda_{i_n}}
$$
$$
= 1 + \sum_{n=1}^{\infty} \frac{1}{n!}(\kappa - \mathrm{i}s - u)^n \left(\mathrm{Tr}\left(\frac{1}{u - H}\right)\right)^n_{\mathrm{smooth}},
$$
$$(136)$$

where the subscript on the products of resolvents means we subtract contact terms. The simplest cases are

$$
\mathrm{Tr}\left(\frac{1}{u - H}\right)_{\mathrm{smooth}} = \mathrm{Tr}\,\frac{1}{u - H}, \quad \left(\mathrm{Tr}\left(\frac{1}{u - H}\right)\right)^2_{\mathrm{smooth}} = \left(\mathrm{Tr}\left(\frac{1}{u - H}\right)\right)^2 - \mathrm{Tr}\left(\frac{1}{u - H}\right)^2.
$$
$$(137)$$

Now one can take the discontinuity for each term in the expansion (136). The constant in (136) does not contribute, since the pole at $\kappa - is$ lies outside of the contour. Using Sokhotski-Plemelj, we obtain

$$
\rho_{H_0}(\kappa) = \frac{g^2}{2\pi} \int_{-\infty}^{+\infty} ds \, \exp\left(-\frac{g^2}{2}s^2\right) \int_{-\infty}^{+\infty} du \, \exp\left(-\frac{g^2}{2}(\kappa-u)^2\right) \times
$$
$$
\times \sum_{n=0}^{\infty} \frac{1}{n!}(\kappa - is - u)^n \left(\text{Tr}\,\delta(u-H)\left(\text{Tr}\left(\frac{1}{u-H}\right)\right)^n\right)_{\text{smooth}},
$$
(138)

where again the smooth quantities are defined by subtracting diagonal contact terms. For example

$$
\left(\text{Tr}\,\delta(u-H)\,\text{Tr}\left(\frac{1}{u-H}\right)\right)_{\text{smooth}} = \text{Tr}\,\delta(u-H)\,\text{Tr}\left(\frac{1}{u-H}\right) - \text{Tr}\left(\delta(u-H)\frac{1}{u-H}\right). \quad (139)
$$

Notice that there are no products of delta functions, because the sum in (136) is over different eigenvalues. Finally, the integrals over $s$ give $n$-th order Hermite polynomials, which, by using the Rodrigues formula, can be converted in derivatives of the Gaussian centered at $u = \kappa$. After $n$ partial integration, we then arrive at

$$
\rho_{H_0}(\kappa) = \frac{g}{(2\pi)^{1/2}} \int_{-\infty}^{+\infty} du \, \exp\left(-\frac{g^2}{2}(\kappa-u)^2\right) \times \quad (140)
$$
$$
\times \sum_{n=0}^{\infty} \frac{1}{n!}\frac{(-1)^n}{g^{2n}} \partial_u^n \left(\text{Tr}\,\delta(u-H)\left(\text{Tr}\left(\frac{1}{u-H}\right)\right)^n\right)_{\text{smooth}}
$$
$$
= \frac{g}{(2\pi)^{1/2}} \int_{-\infty}^{+\infty} du \, \exp\left(-\frac{g^2}{2}(\kappa-u)^2\right) (\text{Tr}\,\delta(u-H) + \dots). \quad (141)
$$

Each term in this expansion has a direct gravitational interpretation, which we will discuss next.

The leading contribution represents the insertion of a spectral density operator $\text{Tr}\,\delta(\kappa-H)$ in the $H$ integral, and in gravity this corresponds to inserting one extra asymptotic boundary, with fixed energy boundary conditions. The difference with $g = \infty$, is that the energy of the boundary state will be smeared with a tight Gaussian. The leading effect on the eigenvalue correlation (132) is a similar smearing

$$
T_\kappa(E_1, E_2) = (S(E_1, E_2) - S(E_1, \kappa)S(E_2, \kappa)/S(\kappa, \kappa))^2 \quad (142)
$$
$$
+ \frac{g^2}{2\pi} \exp\left(-\frac{g^2}{2}(E_1 - \kappa)^2 - \frac{g^2}{2}(E_2 - \kappa)^2\right). \quad (143)
$$

Close to the almost-fixed eigenvalue this is indistinguishable from the results for the finite dimensional matrix integral (42), but here with a clear gravitational interpretation. See also Fig. 11.

The subleading corrections to (141) correspond with having multiple extra boundaries. They come in two types. The first one is the contact term contributions and has segments with different boundary condition separated by marked points [56, 72, 107, 108]. The second type is simply the coming from the multi-trace contributions in the first line of (141). Both are shown in Fig. 13.

The partial derivative $\partial_u$ introduces an extra marked point on any of the boundaries [56, 72, 107, 108]; fundamentally the boundary conditions remain the same.

$$
\cdots \; = \; \int_{-\infty}^{+\infty} du \exp\left(-\frac{g^2}{2}(\kappa-u)^2\right) \cdots
$$

$$
- \frac{1}{g^2}\int_{-\infty}^{+\infty} du \exp\left(-\frac{g^2}{2}(\kappa-u)^2\right) \cdots \cdots \; + \; \cdots
$$

Figure 14: Gravitational interpretation (right) for inserting $\rho_{\mathsf{H}_0}(\kappa)$ (left) in our two-matrix integral (126). The eigenbrane boundary is smeared away from the $g=0$ limit.

In summary, we end up with the mapping of fixed energy boundaries for the auxiliary $\mathsf{H}_0$ matrix, to tightly smeared gravitational boundary conditions, that is shown in Fig. 14.

Notice that, when lowering $g$ to make the eigenvalue more random, the configurations with many macroscopic boundaries are no longer suppressed and ultimately they proliferate. This reminds us of the tearing phenomenon encountered in section 4, but now approached from the small $\sigma$ regime. It is surprising that in this setup, the gravitational theory seems to make more sense when one eigenvalue is completely fixed, than when the eigenvalue is *half*-random.

This provides hope for the endpoint $\sigma=0$ of the theory, where we try fixing the whole Hamiltonian (1). Perhaps when lowering from $\sigma=\infty$, the theory goes through some rough patch at intermediate values of $\sigma$ where spacetime appears to be broken, torn apart by macroscopic holes, but then regains its footings and acquires a nice gravitational interpretation again at $\sigma=0$.

## 6 Concluding remarks

We have investigated the matrix integral

$$
\mathcal{Z}(\sigma, \mathsf{H}_0) = \int dH \, \exp\left(-L\operatorname{Tr}V(H) - \frac{L}{2\sigma^2}\operatorname{Tr}(\mathsf{H}_0-H)^2\right), \tag{144}
$$

in different parametric regimes of $\sigma$, both in finite dimensional matrix integrals and in the double-scaling limit, where the theory describes two dimensional dilaton gravity. This represent a more realistic toy model for higher dimensional quantum gravity, which appears to be dual to a single boundary theory, instead of an ensemble like JT gravity.

Our most important findings are:

1. Wormholes gradually approach diagonal delta functions in the non-random theory.

2. One universal saddle $S=0$ in the Efetov model governs the non-averaged theory.

3. When making the theory less random, there are phase transitions where spacetime is torn apart.

It has been suggested that perhaps quantum gravity is just an ensemble average, and that is the end. However, via wormhole physics, traces of microstructure have been discovered

within gravitational systems, like the ramp and plateau. Analogous to how Brownian motion was evidence for molecules, this is evidence that there *is* microstructure underlying spacetime. The logical next step is to investigate what the atoms of spacetime are. Our work is a step in that direction.

We refer to the individual sections and the summary in section 1.2, for specific discussions on each regime. We end this work with various, more speculative, pieces of discussion and raise open questions.

### Higher dimensions

Unlike with two or three dimensions, quantum gravity in higher dimensional AdS is dual to one single boundary theory. From our analysis, we have learned what it means, for a two dimensional theory, to go towards a single boundary theory. Importantly, we saw in section 3 that there are perfectly sensible theories of dilaton gravity (116) which are less random than the simplest case of JT gravity.

This confirms the idea that, when we consider a UV complete theory of quantum gravity, which we believe are rather scarce and special; the UV details of the theory, such as branes, strings, higher-spin fields etcetera, are encoded in specific couplings of the effective low energy bulk description. For many questions, however, a truncation to the Einstein-Hilbert or JT action suffices. But for questions about, say, factorisation [90,109] it does not. The simplified gravity theory appears to be dual to an ensemble. It is the additional bulk couplings (that we dropped in doing the truncation) that then need to be taken into account. Our model precisely shows that when we move away from the boundary theory being an ensemble, bulk couplings appear and in particular they depend heavily on the specific boundary theory. This also highlights the point that one specific boundary theory is dual to one specific bulk theory.

It would be interesting to study our deformed JT gravity theory in Lorentzian signature. The extra boundaries labelled by $x_i$ would then, after analytic continuation, correspond to additional boundaries in Lorentzian spacetime, seemingly just outside the horizon like fuzzballs [68]. Do these micro-structures also generalise to higher dimensions? If and how these structures relate to microstates of black holes is an interesting question and requires a full understanding of the theory at small $\sigma$, which seems unclear at present.

The most promising avenue towards understanding small $\sigma$, seems to be understanding the $S = 0$ universal saddle of the Efetov model in gravity.

Of course, there is an alternative open-closed dual Lorentzian interpretation of literally JT gravity with a deformed dilaton potential (116). It would be interesting to understand the closed dual of the tearing phase. Perhaps this is related to the aforementioned non-monoticities that appear in the dilaton potential.

### Weingarten corrections

The conclusion of sections 4 was that spacetimes are annihilated by the nucleation of huge holes, when $\sigma$ is lowered below some critical value.

However one should remember that the starting point (82) of our analysis is an approximation too, and that approximation comes into jeopardy when the coupling constants blow up. When operators $\mathcal{O}_n$ with huge valence $n$ become relevant, the Gaussian approximation to the Weingarten functions breaks down. This is because all Weingarten functions at fixed $n$, share the same denominator [85,86], which diverges when $n > L$. For example when $n = 3$

$$\text{Wg}(1,1,1) = \frac{L^2 - 2}{L(L^2 - 1)(L^2 - 4)}, \tag{145}$$

the Weingarten functions diverge if $L < 3$. Deviations of Weingarten functions from Gaussian behavior are intimately connected with various signatures of discreteness, such as the plateau. These corrections should be important for a full understanding of small $\sigma$. Furthermore, the combinatorial prefactors in (81) could make the multi-trace deformations compete with the single-trace deformations.

Therefore, we believe a rigorous treatment of the transition to small $\sigma$, will require control of the full Harish-Chandra integral (78) in the double scaling limit; we have not succeeded in understanding this. Undoubtedly this would result in an expression for the deformed potential involving multi-trace deformations on top of the $L$ branes we already had. One would have to figure out how to process this through something like (88) or through the string equation machinery.

## Multitrace deformations and branched polymers

Fortunately, multitrace deformations of matrix models have been considered before and several interesting phenomena were found [110–114].

In [110], a quartic matrix integral deformed by the double trace term $(\mathrm{tr}\,H^2)^2$ is considered. This interaction is known as a *touching* interaction, because in the ribbon graph, ribbons would be touching. If one considers a term $(\mathrm{tr}\,H^k)^2$ then higher interaction vertices are touching, establishing a microscopic wormhole. In the continuum limit, these become nonlocal interactions between distinct points on the spacetime, so one obtains a non-local dilaton gravity action (7). The open string dual are the brane interactions discussed in section 3.4.

As function of the coupling $g$ of these multitrace operators, three phases were found. Below some critical coupling $g_0$ the theory behaves like the standard minimal string, but with nonlocal interactions. Then there is a peculiar phase at $g_0$ where we still have the minimal string, but mysteriously the minimal matter primaries are dressed by the dual Liouville primary with weight $Q - \alpha$ instead of $\alpha$ [113]. For $g > g_0$ the theory is dominated by branched polymers, which seems to signal a breakdown of continuum geometry.

It would be interesting to understand these phases in detail in the context of our finite $\sigma$ theory, in particular one would like to analytically track the non-localities in the dilaton gravity action, and try to make sense of the branched polymer phase in gravity.

## Averaging over bulk couplings

Let us mention that by using a Hubbard-Stratonovich transformation, the double-trace deformation can also be interpreted as a single trace term with Gaussian measure. The microscopic wormholes then originate from an average over bulk couplings, just as Coleman envisioned [115]. This now corresponds in dilaton gravity with viewing the nonlocal theory discussed above, as a local dilaton gravity theory with specific couplings, and with en ensemble average over the couplings. This would be a closed string picture of the effects of branes and their interactions. It is tantalizing that averages over bulk couplings appear when we are trying to describe the bulk dual of one system. The idea would be that this ensemble too ultimately collapses when $\sigma = 0$, then we are in an $\alpha$-state [9, 12].

We have seen that introducing the external matrix $H_0$ generates bulk couplings, as manifested in the deformed dilaton potential (116). From the matrix model perspective, we have

$$
\begin{aligned}
\int \mathrm{d}\mathsf{H}_0 \, \mathcal{Z}(\sigma, \mathsf{H}_0) &= \int \mathrm{d}H \, \mathrm{d}\mathsf{H}_0 \, \exp\left(-L\,\mathrm{Tr}\,V(H) + \frac{1}{2\sigma^2}\,\mathrm{Tr}(\mathsf{H}_0 - H)^2\right) \\
&= \int \mathrm{d}H \, \exp\left(-L\,\mathrm{Tr}\,V(H)\right)
\end{aligned}
\tag{146}
$$

modulo implicit normalization constants. An interesting open problem is understanding why averaging over $H_0$ in the closed string description (116) returns simply JT gravity, without any matrix technology. One way to understand this, would be to find a gravitational interpretation for the other poles in the dispersion relation (68), and for them vanishing when we integrate over $H_0$.

Another place where averaging over bulk couplings appeared was in section 5. There we considered fixing only one eigenvalue of $H$ and needed to integrate over the brane parameters. Directly interpreting (135) as averaging over brane locations is, however, subtle; because determinants and branes differ by a factor $\exp(-LV(E)/2)$. This diverges in the double scaling limit, therefore complicating an immediate gravitational interpretation of (135). It would be interesting to understand how to deal with this, such that one could study (135) away from small $\sigma$ perturbation theory.

### Direct product of gravity theories

From the matrix integral point of view, the genus zero spectral density now has many cuts (66). Perhaps for sufficiently small $\sigma$, one could interpret the matrix integral as a direct product of $L$ gravity theories, which only know about each other non-perturbatively (see also [116]). Thus perhaps there is some many-universe interpretation [9] at small $\sigma$.

One way to also see that this could be true is by looking at the topological recursion for matrix models with an external field [47,117]. This recursion (and hence also the topological expansion) is much more complicated then in the usual case, not only because the spectral curve is more intricate, but also, and perhaps most importantly, because the residue is not just taken at $z = 0$ (as is the case for JT for instance), but at all branch points of the spectral curve (spectral edges). Since there are many of them, the topological recursion includes many more contributions. In the naive double scaled theory one could argue that only one branch point is of interest, but clearly at small $\sigma$ this is insufficient.

### Open questions

We have made progress in understanding non-averaged two dimensional gravity. However, many open questions remain. There are several concrete things to investigate:

1. Gravitational interpretation for the universal $S = 0$ saddle in the Efetov model.

2. Double scaling limit of the Efetov model (54). Gravitational interpretation for the theory whose spectrum consists of many tight semicircles, centered around each of the target eigenvalues (66). Investigate the leading order wormhole for that theory.

3. Gravitational interpretation for the residues from the other poles in the dispersion relation (68).

4. Solve matrix integrals with multi-trace deformations in the potential. Gravitational interpretation of the corresponding double scaling limit, resulting in a concrete nonlocal dilaton gravity action. Some progress in this direction has been made in [110–114].

5. Describe the atoms of non-averaged gravity at $\sigma = 0$.

### Acknowledgments

We thank Alex Belin, Jan de Boer, Raghu Mahajan, Thomas Mertens, Onkar Parrikar, Steve Shenker, Douglas Stanford, Misha Usatyuk, Zhenbin Yang and Shunyu Yao for stimulating discussions. AB is a BAEF fellow and is also supported by the SITP. JK is supported by the Simons Foundation.

## A  Efetov saddle points

In this appendix, we collect some details about the saddle point structure of the Efetov model for one single determinant (56); in a simple example where $\mathsf{H}_0$ has $L/2$ eigenvalues $z$ and $L/2$ eigenvalues $-z$. In this case the saddle point equation becomes

$$\frac{8}{a^2}\mathrm{i}S = \frac{1}{E - za^2/4\sigma^2 - \mathrm{i}S} + \frac{1}{E + za^2/4\sigma^2 - \mathrm{i}S}\,. \tag{147}$$

This is a cubic equation for $S$ and can be solved analytically, but the solutions are a bit unwieldy, so we resort to a numerical analysis. The basic things we want to highlight are the movement of the solutions as a function of $\sigma$, which we sketched in Fig. 15 for two different energies. One energy remains outside of all cuts and the other enters and leaves a cut as sigma decreases. The discussion is in the caption of Fig. 15.

The question is which of these saddle points lies on the integration contour. This quickly becomes teadious to answer. Fortunately, we have made some educated guesses in section 2.3. We can simply check if these are correct by computing (56) numerically and comparing it to the saddle point approximation, where we take only the physical saddle into account for energies outside any cut; and take the physical saddle plus the saddle with the opposite branch for the relevant square root, whenever we are inside some spectral cut. We find excellent agreement,

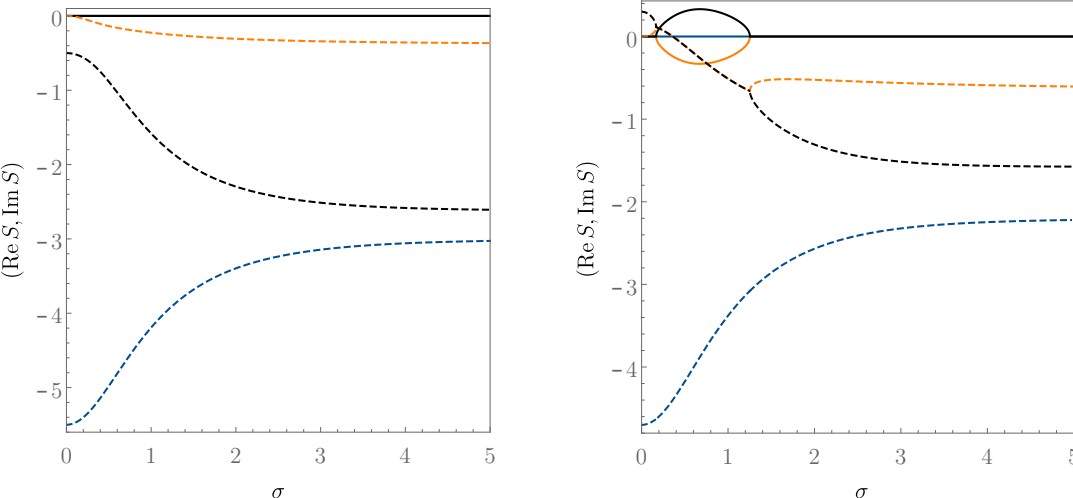

Figure 15: Three solutions (orange, black, blue) to (147) as a function of $\sigma$. The real part of $S$ is in solid lines and the imaginary part of $S$ is in dashed lines. We use $z = 5/2$, $b = 2$ and $E = 3$ for the left figure and $E = 2.2$ for the right. At large $\sigma$ the physical saddle (orange) should approach $-\mathrm{i}/2(E - \sqrt{E^2 - b^2}) \approx -0.382\mathrm{i}$ for the left and $-0.642\mathrm{i}$ for the right plot, as can be seen from the plots. Notice that the physical saddle approaches $S = 0$ for $\sigma = 0$ as claimed in the main text, whereas the other saddles indeed approach $S = -\mathrm{i}(E \pm z)$. Near $\sigma = \infty$ one saddle (blue) approaches the value $S = -\mathrm{i}E$ corresponding with (64) and another (black) approaches $-\mathrm{i}/2(E + \sqrt{E^2 - b^2})$, the other standard solution in (62). For the right plot, we can see that $E = 2.2$ enters the cut when the physical saddle and one of the other saddles coincide, it leaves the cut again at the second bifurcation. Both these transitions take place at an (anti-)Stokes line, as claimed in the main text, since both the real and imaginary parts of the saddles coincide. In the region between the two bifurcations, both saddles are on the integration contour, otherwise only the physical one is included (orange).

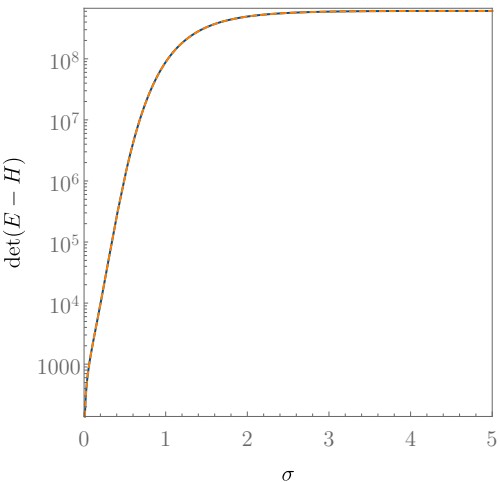
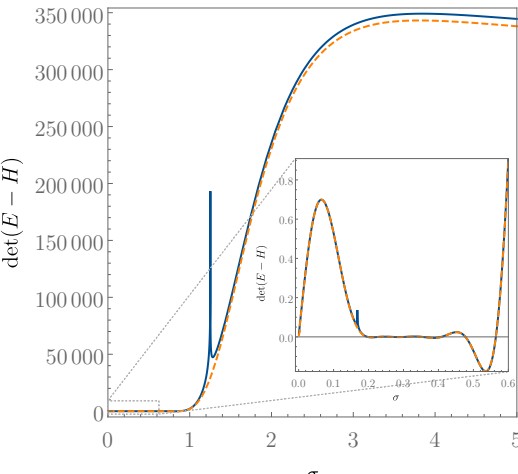

Figure 16: Comparison between numerical evaluation of (56) (orange dashed) and its saddle point approximation (solid blue) at $L = 20$, taking only the saddles mentioned in the text into account. We used $b = 2$ and $z = 5/2$. The plot on the left is a log plot (because the range is rather big) for $E = 3$ which remains outside any cut as a function of $\sigma$, and we see that the saddle point approximation is excellent. For the right figure we have $E = 2.2$ and will thus enter and leave a cut as $\sigma$ decreases. The saddle point approximation is still very good, except when the value of $E$ enters a cut around $\sigma \approx 0.16$ and $\approx 1.2$. At these values for $\sigma$ the saddles change dominance and the saddle point approximation breaks down, leading to bigger errors. Notably, between those two values there are two saddles (orange and black in Fig. 15) contributing. The inset on the right shows the same plot, but for smaller values of $\sigma$.

as shown in Fig. 16.[15]

We expect this to be true more generally, but it would be worthwhile to verify it more analytically, by computing steepest descent contours etcetera.

## B Quartic matrix integral

In this appendix we study some aspects of a quartic matrix model and its double scaling. In particular, we will present a straightforward way of obtaining an $E^{3/2}$ edge and one that includes a $E^{1/2}$ edge as well. The latter is what is encountered in the $(2, 3)$ minimal string, also known as pure gravity; however in the current context that name is misleading, all minimal strings are pure dilaton gravity [5, 56–59].

We consider the potential

$$V(H) = \frac{1}{2t}H^2 - \frac{t_4}{4t}H^4, \tag{148}$$

and we will scale to the critical point corresponding with the $(2, 3)$ minimal string. This model is simple enough to be didactic, and sufficiently rich to clarify the intricacies of double scaling to anything except the $(2, 1)$ minimal string; which you obtain everywhere except at the critical points.

With the main sections in mind we will include the quadratic deformation from (82), but

---

[15] Including another saddle given an answer that is many orders of magnitude too large to match (56).

leave out the inverse determinants; and thus study the potential

$$W(H) = \left(\frac{1}{t} + \frac{1}{\sigma^2}\right)\frac{1}{2}H^2 - \frac{t_4}{t}\frac{1}{4}H^4 = \frac{1}{2\tau}H^2 - \frac{\tau_4}{4\tau}H^4, \tag{149}$$

where we introduced new effective coupling constants, similar to in (9)

$$\frac{1}{\tau} = \frac{1}{t} + \frac{1}{\sigma^2}, \quad \frac{\tau_4}{\tau} = \frac{t_4}{t}. \tag{150}$$

Let us now compute the resolvent using (88). We deform the integration contour around the pole at $E$ and the pole at $\infty$, which, in the latter case, should be computed by going to variables $\lambda = 1/z$. The one at $E$ gives $W'(H)/2$, and does not contribute to the discontinuity of the resolvent; and therefore neither to the spectral density. The pole at $\infty$ does give an interesting contribution, it reads

$$R(E) = -\frac{L}{2\tau}(E^2 - E_0^2)^{1/2}\left(1 - \frac{3}{2}\tau_4 E_0^2\right) + \frac{L\tau_4}{2\tau}(E^2 - E_0^2)^{3/2}, \tag{151}$$

resulting in the spectral density

$$\rho(E) = \frac{L}{2\pi}\frac{1}{\tau}(E_0^2 - E^2)^{1/2}\left(1 - \frac{3}{2}\tau_4 E_0^2\right) + \frac{L}{2\pi}\frac{\tau_4}{\tau}(E_0^2 - E^2)^{3/2}. \tag{152}$$

We want to think about the constraint (86) as fixing the parameter $\tau$ as a function of $E_0$, such that there is the freedom to send $E_0$ to infinity for double scaling

$$\tau = \frac{E_0^2}{4}\left(1 - \frac{3}{4}\tau_4 E_0^2\right), \tag{153}$$

which reduces indeed to the Gaussian potential (9), when we turn off the quartic term. This equation eliminates $t$ when translated back to the original couplings (149).

To double scale this theory we send $E_0$ to infinity and considers energies close to the spectral edge (97), whilst simultaneously sending $L$ to infinity; in such a way that the spectrum near the edge remains finite. We believe it is didactic to work this out in some detail. It seems sensible to scale $\tau_4$ as $\tau_4 = g/E_0^2$ with $g$ finite, giving

$$\rho(E) \overset{\text{ds}}{=} \frac{L}{\pi}\frac{2^{3/2}}{E_0^{3/2}}\frac{1 - 3g/2}{1 - 3g/4}E^{1/2} - \frac{L}{\sqrt{2}\pi}\frac{1}{E_0^{5/2}}\frac{1 - 19g/2}{1 - 3g/4}E^{3/2}. \tag{154}$$

The second term is suppressed by $1/E_0$ for generic coupling. We are then urged to scale $L = e^{S_0}(E_0/2)^{3/2}$ to obtain some finite answer near the edge

$$\rho(E) = \frac{e^{S_0}}{\pi}\frac{1 - 3g/2}{1 - 3g/4}E^{1/2}. \tag{155}$$

This is the spectral curve for the $(2,1)$ minimal string, or topological gravity [5,118]. Generic potentials indeed always double scale to this simplest $(2,1)$ minimal string.

To obtain the $(2,p)$ minimal strings one should tune (in the quartic case) the couplings of the potential such that the coefficient of the $E^{1/2}$ vanishes, making the $E^{3/2}$ term competitive. For $p = 2m + 1$, the couplings multiplying $H^{2+2m}$ are tuned to make the first $m$ terms in the expansion vanish, leaving only $E^{m+1/2}$, these special couplings are called critical points. In our case we must take $g = 2/3$, commonly written as; after using (153)

$$\tau_4 = \frac{1}{12\tau}. \tag{156}$$

Since the leading density is being tuned to zero, we need much more eigenvalues $L$ in the theory to see interesting behavior near the edge – from (154) we see that we should take $L = e^{S_0}(E_0/2)^{5/2}$, and find

$$\rho(E) = \frac{e^{S_0}}{\pi} \frac{4}{3} E^{3/2}. \tag{157}$$

This is indeed the spectral curve of the second critical point. We can make the lower terms competitive at the same order, by scaling slightly differently towards the critical points. For this quartic example, choosing the coupling $g = 2/3(1 - 2\kappa/E_0)$ gives

$$\rho(E) = \frac{2e^{S_0}}{\pi} \left( \kappa E^{1/2} + \frac{2}{3} E^{3/2} \right). \tag{158}$$

This spectral density is indeed proportional to the spectral density of the $(2, p)$ minimal string theory with non-zero cosmological constant $\kappa$, with $p = 3$ [5]

$$\rho(E) \propto e^{S_0} \sinh \left[ \frac{p}{2} \operatorname{arccosh} \left( 1 + \frac{E}{\kappa} \right) \right]. \tag{159}$$

As we send the cosmological constant to zero, one then indeed recovers the second multi-critical point (157).

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
