# Peer review of "Gravity without averaging"

_SciPost Physics, doi:SciPost Phys. 12, 073 (2022)_

## Round 1 · Author Response

Dear Editor,

We thank the referees for their thorough reading and useful and interesting comments and feedback. We also notice that 3.39 and 3.40 contained an error and so we changed it. We also added a figure 10 to clarify the tearing transition a bit more. These were outside the questions/comments of the referees. See below for our reply to their feedback.

Referee 1:

1) We thank the referee for this comment. The referee is correct that the map from the genus zero spectral density to the dilaton potential is not very simple. The result we have written down here is valid up to order $e^{-S_0}$ as written above 3.39. To emphasize this more, we have added to equations 3.40 and 3.39 and $+ \mathcal{O}(e^{-S_0})$.

The phrase 'less random' here means that we have done perturbation theory in $1/\sigma^2$ around the $\sigma = \infty$ point (and in the double scaling limit). We do not have an understanding of what the precise dilaton gravity is (if it exists) when $\sigma$ is finite (but potentially still large), partly because of the discussion in section 4.

2) We have not pursued a thorough classical analysis of the dilaton gravity with potential 3.39, because we felt the paper was already rather long and had many interesting results. That said, it is certainly interesting to study. The large $\Phi$ asymptotics of the potential are dominated by the $2\Phi$ term in $W$ as $U$ is exponentially decaying in that direction.

3) This is a very interesting question and we have thought about it, but have not understood what it implies in Lorentzian signature.

4) Another great question. The referee is correct as the JT gravity spacetime arises from the worldsheet of the $(2,p)$ minimal string theories. What we had in mind here is that perhaps a dimensional reduction of the higher dimensional fuzzball to 2d would give rise to the branes we encountered in JT gravity, but we have not made any efforts to try to make that more precise.

Referee 2:

1) To address this point of the referee we have changed the abstract so that it makes clear what we study and what our results are. We have in particular added 'aspects of' in the first sentence, as to emphasize that we do not solve the whole problem, which we were never trying to claim. We have also added the sentence 'a good gravitational description at small values of sigma remains elusive' to further drive home this point.

2) We thank the referee for this comment. The manuscript already contains a rather extensive summary section 1.2 and without further specification of the referee what is unclear we cannot really do something more here.

3) We thank the referee for the question. The subscript $H_0$ on the contour integral is supposed to mean that we take a small contour around each eigenvalue $x_i$ of $H_0$. We have clarified this again below 2.14.

4) We have changed this by replacing the more colloquial words.

5) See point 2.

Referee 3:

1) We thank the referee for the interest in this comment. Formula (5.8) represents the degree to which the amplitude does not factorize, in particular the second term. Locally near each fixed eigenvalue there is a quadratic zero. When $E_1$ or $E_2$ is more then a few $e^{-S_0}$ away from $\kappa$, the effect of the fixing is shielded by the remaining random eigenvalues, see Figure 11.

2) We thank the referee for this interesting comment. The auxiliary holes would indeed need to carry some information about the $H_0$ eigenvalues. However it is not clear to us why all geometries ending on such holes would give precisely zero when averaging over $H_0$.

3) We thank the referee for spotting the typos, we have changed them.

---

## Editorial Decision

published